# Sustained micellar delivery via inducible transitions in nanostructure morphology

Nicholas B. Karabin[1], Sean Allen[2], Ha-Kyung Kwon[3], Sharan Bobbala[1], Emre Firlar[4,5], Tolou Shokuhfar[4], Kenneth R. Shull[3] & Evan A. Scott [ID] [1,2,6,7,8]

Nanocarrier administration has primarily been restricted to intermittent bolus injections with limited available options for sustained delivery in vivo. Here, we demonstrate that cylinder-to-sphere transitions of self-assembled filomicelle (FM) scaffolds can be employed for sustained delivery of monodisperse micellar nanocarriers with improved bioresorptive capacity and modularity for customization. Modular assembly of FMs from diverse block copolymer (BCP) chemistries allows in situ gelation into hydrogel scaffolds following subcutaneous injection into mice. Upon photo-oxidation or physiological oxidation, molecular payloads within FMs transfer to micellar vehicles during the morphological transition, as verified in vitro by electron microscopy and in vivo by flow cytometry. FMs composed of multiple distinct BCP fluorescent conjugates permit multimodal analysis of the scaffold's non-inflammatory bioresorption and micellar delivery to immune cell populations for one month. These scaffolds exhibit highly efficient bioresorption wherein all components participate in retention and transport of therapeutics, presenting previously unexplored mechanisms for controlled nanocarrier delivery.

[1] Department of Biomedical Engineering, Northwestern University, 2145 Sheridan Road, Evanston, IL 60208, USA. [2] Interdisciplinary Biological Sciences, Northwestern University, 2205 Tech Drive, Evanston, IL 60208, USA. [3] Department of Materials Science and Engineering, Northwestern University, 2220 Campus Drive, Evanston, IL 60208, USA. [4] Department of Bioengineering, University of Illinois at Chicago, 851 South Morgan Street, Chicago, IL 60607, USA. [5] Department of Mechanical and Industrial Engineering, University of Illinois at Chicago, 842 West Taylor Street, Chicago, IL 60607, USA. [6] Chemistry of Life Processes Institute, Northwestern University, 2170 Campus Drive, Evanston, IL 60208, USA. [7] Simpson Querrey Institute, Northwestern University, 303 East Superior Street, Chicago, IL 60611, USA. [8] Robert H. Lurie Comprehensive Cancer Center, Northwestern University, 303 East Superior Street, Chicago, IL 60611, USA. Correspondence and requests for materials should be addressed to E.A.S. (email: evan.scott@northwestern.edu)

Nanocarriers present a versatile means of delivering therapeutic and diagnostic agents to specific cells and tissues and have become a primary basis for theranostic strategies[1–6]. Targeted nanocarrier delivery systems are primarily administered by bolus intermittent injections and infusions, with few options available for sustained delivery[7]. Sustained delivery platforms have proven highly advantageous for drug administration, particularly for long-term processes such as wound healing[8,9], hormone therapy[10,11], and transplant tolerance[12,13], and incorporation of nanocarriers may further improve the efficacy and flexibility of these applications[14,15]. Due to the physicochemical similarities and cellular biodistributions between nanoscale materials and viruses, nanocarriers have demonstrated considerable advantages for the targeting and modulation of phagocytic antigen-presenting cells (APCs) that are critical for eliciting immune responses during vaccination and immunotherapy[1,2,16,17]. As a result, controlled long term delivery of nanocarriers may present avenues for immunization and the treatment of diseases characterized by severe immune dysregulation, such as cancer, cardiovascular disease, and diabetes[1,2].

The most common sustained nanocarrier delivery platforms are composite systems that rely on implanted or injected hydrogel networks to entrap the nanoparticulate delivery vehicles[18–22]. Such systems employ either crosslinked synthetic or natural biopolymers to modulate the diffusive release of entrapped nanocarriers, but given the structural role of the polymer network, only a fraction of the total material present in the construct plays a direct role in delivery of the active. Therefore the bulk of the hydrogel material serves no direct therapeutic purpose and may instead elicit chronic inflammatory responses with or without controlled degradation[23,24]. The primary disadvantage of these hydrogels is the foreign body response, which eventually leads to isolation of the implant through formation of a fibrous capsule that can induce patient discomfort and disruption of nanocarrier release kinetics[23,25]. To enhance tolerability of nanocarrier-loaded hydrogels, alternative strategies have emerged that include hydrogels composed of non-covalently linked nanocarriers themselves without polymer matrices[26], as well as hydrogels composed of polymers with reduced inflammatory potential[24,27]. We sought to combine and improve upon both these strategies by developing a synthetic macromolecular hydrogel network that could dynamically restructure into monodisperse nanoscale vehicles for non-inflammatory bioresporption and sustained nanocarrier delivery.

Depending on the method of preparation and solution conditions, block copolymer (BCP) systems can controllably self-assemble into nonequilibrium structures that can be induced to further transition into different thermodynamically stable morphologies upon appropriate stimulation[28–31]. For example, BCPs can assemble into high aspect ratio cylindrical filomicelles (FMs) that can transition to spherical micelles (MCs) under a variety of conditions[30–35]. Cryogenic transmission electron microscopy (cryoTEM) has been used to capture various snapshots of these processes[30,33–35]. Surface tension-dependent mechanisms at the solvent/BCP interface for these cylinder-to-sphere (i.e., FM-to-MC) transitions have been investigated both empirically and theoretically[30,31,33–35]. Since BCPs have been employed for the formation of hydrogels[36–38], we hypothesized that a hydrogel composed of FMs that are susceptible to inducible or continuous changes in surface tension may be employed for sustained micellar delivery. As the cylindrical FMs transition to their spherical counterparts, the primary structural component of the hydrogel depot would become an active participant in delivery of therapeutic or diagnostic payloads.

Poly(ethylene glycol)-bl-poly(propylene sulfide) (PEG-bl-PPS) is a versatile BCP system that has been utilized to produce a variety of self-assembled nanocarriers, many of which can undergo oxidation-dependent changes in nanostructure[39–41]. The assembled nanostructure is a function of the PEG-bl-PPS hydrophilic mass fraction (molecular weight ratio of the hydrophilic to hydrophobic copolymer blocks)[39,41–43,44]. By controlling the block lengths of PEG-bl-PPS, monodisperse populations of spherical MCs, vesicular polymersomes, and cylindrical FMs have been produced for the delivery of both hydrophilic and lipophilic payloads[29,43–46]. The ability to form diverse nanostructure morphologies can be partially attributed to the low Tg (227 K) and resulting high chain flexibility of the hydrophobic PPS block that permits rapid transitions between metastable aggregate morphologies[29,47,48]. Importantly, PEG-bl-PPS oxidizes to more hydrophilic poly(propylene sulfoxide) or poly(propylene sulfone) copolymers, which allows rapid and controlled oxidation-triggered transitions of PEG-bl-PPS nanostructure morphologies[39,41]. Oxidation has been previously employed to induce payload release from the aqueous interiors of PEG-bl-PPS polymersomes[39,46,49], but has not been used to induce cylinder-to-sphere transitions for controlled delivery.

We hypothesized that cylinder-to-sphere transitions may occur for PEG-bl-PPS FMs due to changes in surface tension that occur under oxidative conditions as propylene sulfide converts to more hydrophilic derivatives. After characterizing FMs in solution via cryoTEM and small angle X-ray scattering (SAXS), we attempt to capture these morphologic transitions via cryoTEM. Thermodynamic modeling is subsequently used to verify the potential driving force for the cylinder-to-sphere transition. Modification of the PEG-bl-PPS synthesis allows for the generation of surface reactive FMs that can be covalently crosslinked together to form a macroscopic scaffold. The transition of FM-scaffolds to micellar delivery vehicles in response to either photo-oxidation or physiologic oxidation is studied both in vitro and in vivo, respectively. Studies are conducted to characterize the released MCs and confirm their presence in vivo. Lastly, the sustained release of micellar vehicles is explored in vivo. Nanostructure uptake by APCs within various lymphoid tissues and histological analysis of the tissue surrounding the injection site are assessed to highlight the platform's potential for future in vivo applications.

## Results

### Characterization of PEG-bl-PPS FM and their transition to MC.

FMs were prepared from methoxy-functionalized PEG$_{45}$-bl-PPS$_{44}$ BCPs (MeO–BCP) (Fig. 1a, Supplementary Fig. 1). CryoTEM confirmed that PEG-bl-PPS BCPs with a hydrophilic mass fraction of 0.38 formed stable FMs exhibiting PPS core radii estimated between 8–10 nm and lengths in excess of a micron (Fig. 1b). SAXS analysis of FM characteristics was achieved by fitting the FM scattering profile with a flexible cylinder model[50] ($\chi^2 = 0.068$) using a cylinder length of 2 μm, a persistence length of 150 nm and a PPS core radius of 8 nm (Fig. 1c), which corresponded well with the observed cryoTEM. We further employed cryoTEM to capture morphologic transitions at the high curvature ends of FMs assembled from MeO-BCPs (Fig. 1d, e, Supplementary Fig. 2). Clusters of MCs were visually confirmed to concentrate primarily at the ends of FMs and were suggestive of sequential release (Fig. 1d, e). Three-dimensional cryoTEM tomography verified that the depicted MCs were not the result of FMs oriented perpendicularly to the sample grid but were in fact a separate morphology (Supplementary Movie 1). The resulting micrographs exhibit similarities to previously described FM end-associated transitions[33,34].

### Thermodynamic modeling of FM-to-MC transition.

Given the oxidation sensitivity of the hydrophobic block and the impact

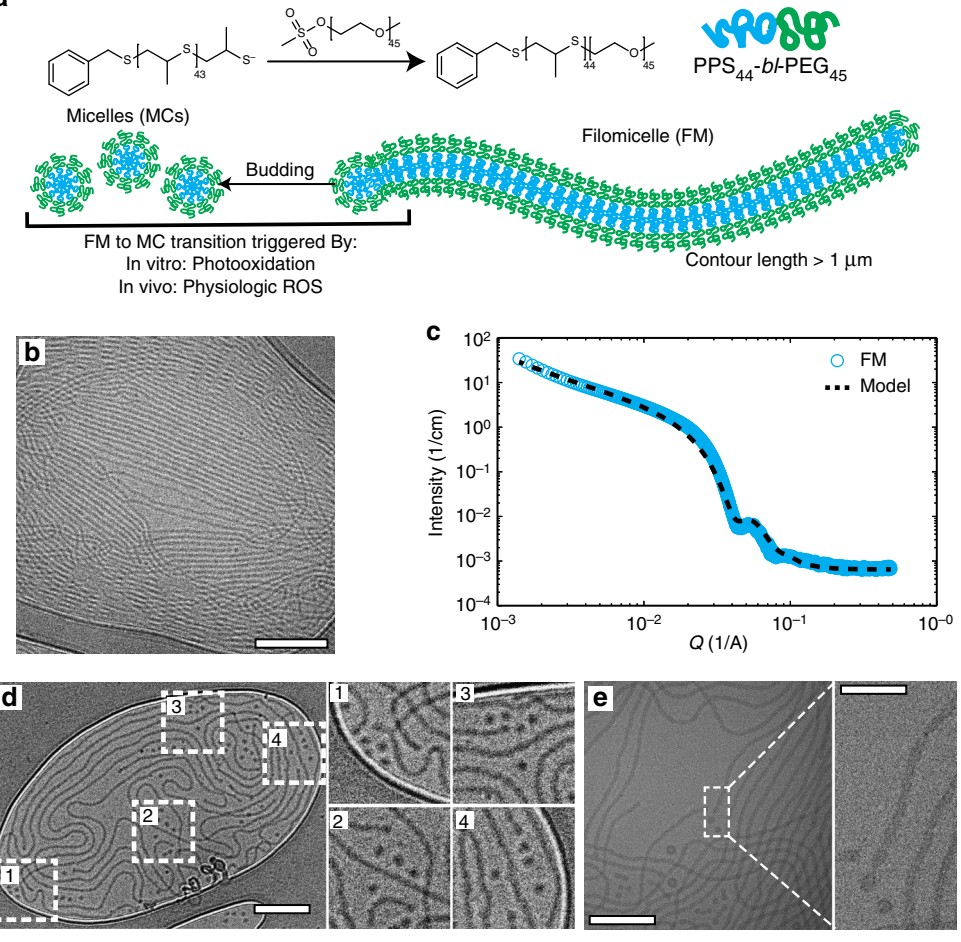

**Fig. 1** Graphical representation of filomicelles and verification of their morphology. **a** Schematic of PEG-bl-PPS block copolymers (BCP) and diagram of a self-assembled filomicelle (FM) transitioning to a micellar vehicle. **b** Cryogenic TEM micrograph of filomicelles. Scale bar represents 200 nm. **c** Small angle X-ray scattering of a filomicelle solution and corresponding model fit (flexible cylinder model, SASView). **d**, **e** Cryogenic TEM micrographs demonstrate that cylinder-to-sphere transitions occur through a budding mechanism at the ends of filomicelles. Scale bars, **d** 250 nm, **e** 200 nm, **e** inset: 50 nm

that this oxidation has on overall BCP hydrophilicity, we hypothesized that oxidizing agents within the immediate FM environment could provide sufficient stimuli to trigger the cylinder-to-sphere transition within the PEG-bl-PPS BCP system. Previous research in amphiphilic BCP MCs has shown that the cylinder-to-sphere morphologic transition can be driven by interfacial tension[32,51,52]. Using a thermodynamic model and interfacial measurements obtained via drop shape apparatus (DSA), we show that the transition in PEG-bl-PPS can be understood by the reduction of interfacial energy upon oxidation of the sulfide group which is balanced by the chain stretching of core and corona blocks. Our thermodynamic analysis of PEG-bl-PPS in solution follows the framework outlined in theoretical work by Zhulina et al. and Lund et al.[32,51]. In this model, the total free energy of a BCP micelle is written as a sum of three components: the interfacial energy between core and solvent, $F_{int}$; the elastic energy of stretching chains in the core, $F_{core}$; and the energy associated with the chains in the corona, $F_{corona}$[52]. Each contribution can be broken down for cylindrical and spherical morphologies:

$$F_{int} = \frac{A_j\gamma}{Pk_BT} \begin{cases} A_j = 4\pi R_c^2 & \text{Spheres} \\ A_j = 2\pi R_c L & \text{Cylinders} \end{cases} \quad (1)$$

Here, $\gamma$ is the interfacial tension, $R_c$ the core radius, $P$ the aggregation number assuming a compact core: $P = 4\pi R_c^3 N_{avo}/(3V_{PPS})$

and $P = \pi R_c^2 L N_{avo}/V_{PPS}$ for spheres and cylinders, respectively. $V_{PPS}$ is the molar volume of PPS block and $N_{avo}$ is Avogadro's number.

$$F_{core} = k_j \frac{R_c^2}{R_{ee}^2} \begin{cases} k_j = \frac{\pi^2}{16} & \text{Spheres} \\ k_j = \frac{3\pi^2}{30} & \text{Cylinders} \end{cases} \quad (2)$$

Where $R_{ee} = N_{PPS}^{1/2} l_{PPS}$ is the unperturbed end-to-end radius of gyration of PPS, the core-forming block.

$$F_{core} = \begin{cases} \frac{\nu C_F R_c}{\sqrt{s}} \ln\left(1 + \frac{l_{PEG}C_H N_{PEG}\left(\frac{s}{l_{PEG}^2}\right)^{(\nu-1)/2\nu}}{\nu R_c}\right) & \text{Spheres} \\ \frac{2C_F R_c}{\sqrt{s}}\left[\left(1 + \frac{(1+\nu)l_{PEG}C_H N_{PEG}\left(\frac{s}{l_{PEG}^2}\right)^{\frac{\nu-1}{2\nu}}}{2\nu R_c}\right)^{\nu/(\nu+1)} - 1\right] & \text{Cylinders} \end{cases} \quad (3)$$

The total free energy of a micelle, $F_{micelle} = F_{int} + F_{core} + F_{corona}$, is minimized with respect to the core radius, $R_c$. The calculations were performed with all

molecular parameters fixed. This included the compositions and molecular weights known from characterizations, while radii of gyration and segment lengths were estimated from previous works[32]. Numerical prefactor parameters, $C_F$ and $C_H$, were taken from the literature[32,51]. The equilibrium core radius $R_c$ corresponding to free energy minima for spherical and cylindrical MCs is shown in Fig. 2a while the micelle free energy calculated at the equilibrium core radius is shown in Fig. 2b. At high interfacial tension, cylindrical MCs are favored. Cylindrical MCs have a smaller core radius at the same interfacial energy, which reduces the elastic energy of the chains in the core. At lower interfacial tension, the system can accommodate a larger interfacial area, which leads to the formation of spherical MCs that minimize the interchain repulsion in the corona. From the thermodynamic model, we predict the transition to occur at $\gamma \cong 8\,mN\,m^{-1}$.

Experimentally measured values of the interfacial tension at the aqueous and organic solvent interface (Fig. 2c) are shown in Fig. 2d for PEG-bl-PPS prior to and post oxidation in phosphate-buffered saline solution (PBS). Interfacial tension measurements were collected using DSA following a previously published procedure[53]. We show that the oxidation of the PPS block leads to a substantial decrease in the interfacial tension. The final interfacial tension ($\gamma \cong 5\,mN\,m^{-1}$) of the oxidized polymer is below our calculated transition point, indicating that the decrease in $\gamma$ can trigger the transition from cylinders to spheres in PEG-bl-PPS MCs as the PPS block is oxidized.

This model provides qualitative confirmation of interfacial tension-driven cylinder-to-sphere transitions in PEG-bl-PPS assembled morphologies. However, a quantitative comparison is not straightforward for several reasons. First, the measured interfacial tension in DSA represents the energy between PPS and PEG blocks across the chloroform-PBS interface (Fig. 2c). This interface contains a lower concentration of polymer than the model interface between the core and the corona block. Second, the thermodynamic model relies on numerical prefactors and long chain statistics, which are not necessarily applicable in the present system of only 44 PPS and 45 PEG repeat units. Further theoretical work would be necessary to more appropriately capture the description of the present system, but this is beyond the scope of our work presented here.

**FM-scaffold preparation and characterization.** To achieve rapid controllable crosslinking of FMs under physiologic conditions, we synthesized vinyl sulfone (VS)-functionalized PEG$_{45}$-bl-PPS$_{44}$ BCP modules (VS-BCP) (Fig. 3a, Supplementary Fig. 1)[54]. VS-BCPs were co-assembled with non-reactive MeO-BCP at different ratios to achieve FMs with controllable levels of VS surface functionalization for subsequent crosslinking. All BCPs were characterized by $^{1}$H NMR and gel permeation chromatography (Supplementary Fig. 3a–c). Having experimentally observed the FM-to-MC transition and theoretically examined how this transition could be driven through oxidation for individual FMs in solution, we proceeded to assess whether this transition could be exploited when FMs were crosslinked into a macroscopic construct. To investigate sustained micellar release from FM scaffolds, we employed both VS-BCP and MeO-BCP to form modular FMs capable of being crosslinked into filamentous hydrogels (Fig. 3b). At concentrations of 100 mg mL$^{-1}$, these FMs

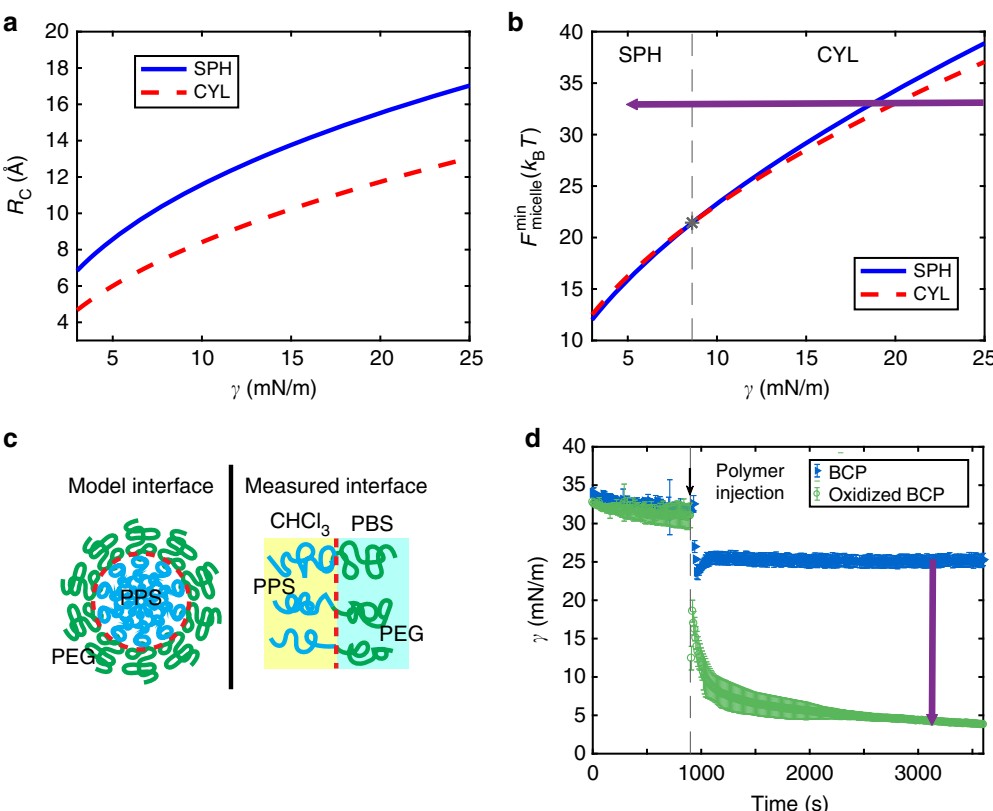

**Fig. 2** Thermodynamic modeling and interfacial measurements of oxidation based cylinder-to-sphere transitions. **a** Equilibrium core radii corresponding to the free energy minima for spherical micelles (SPH) and cylindrical filomicelles (CYL) and **b** total free energy of SPH and CYL calculated at the equilibrium core radii. Star represents the interfacial tension at which cylinder-to-sphere transition occurs ($\gamma \approx 8$). Above this interfacial tension, structures prefer a CYL morphology; below this energy, SPH are formed. As BCP is oxidized, it moves across the map as indicated by purple arrow. **c** Schematic highlighting differences between the modeled and measured systems. **d** Interfacial tension measurements with 2 mg mL$^{-1}$ of BCP injected into the chloroform embedding phase at 900 s. Oxidizing BCP reduces the interfacial tension as indicated by purple arrow

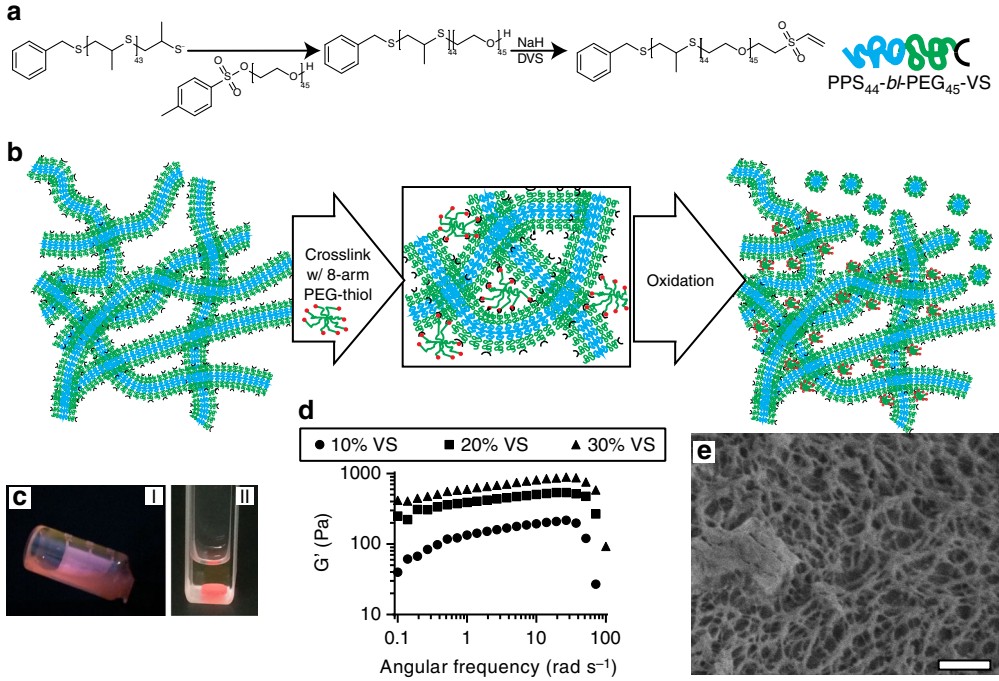

**Fig. 3** Graphical representation and characterization of covalently crosslinked FM-scaffolds. **a** Schematic of vinyl sulfone (VS-BCP) PEG-bl-PPS block copolymer synthesis. **b** Graphical depiction of FM-scaffold crosslinking with eight-arm PEG-thiol and subsequent oxidation-triggered induction of the cylinder-to-sphere transition for release of micelles (MC). **c** Comparison of a 100 mg mL$^{-1}$ modular VS-BCP/MeO-BCP FM solution comprised of 20% w/w VS-BCP before (**I**) and after (**II**) crosslinking into a hydrogel scaffold. **d** Storage moduli of FM-scaffolds obtained via rheometry. **e** CryoSEM micrograph of the underlying fibrous architecture of a FM-scaffold containing 20% w/w VS-PEG-bl-PPS. Scale bar,1 μm

formed viscous solutions that crosslinked into stable hydrogels within minutes in the presence of an eight-arm PEG-thiol crosslinker following injection into molds (Fig. 3c, Supplementary Fig. 4a). Differing the ratios of VS-BCP and MeO-BCP allowed tuning of the rheological properties of the crosslinked hydrogels. Oscillatory mode rheological analysis of the crosslinked scaffolds composed of 10, 20, and 30% of the VS-BCP revealed an increase in elastic modulus over the tested frequency range, as well as a decrease in frequency dependence at low to moderate frequencies (Fig. 3d). Scaffolds exhibited frequency dependence in both their storage and loss moduli at higher frequencies, and the inverse linear dependence of complex viscosity with regard to frequency was indicative of a solid-to-liquid transition (Supplementary Fig. 4c, d). Crosslinked FM-scaffolds demonstrated similar rheological behavior as the physically crosslinked poly(ethylene glycol)-bl-oligo(ethylene sulfide) constructs described previously[55]. Analysis of scaffolds by cryogenic scanning electron microscopy (cryoSEM) revealed an underlying nanoporous architecture with mesh sizes ranging from tens to hundreds of nanometers, reminiscent of collagen matrices and previously reported surfactant and BCP molecular gels (Fig. 3e)[56,57].

**Photo-oxidation of FM-scaffolds for induced release of MCs in vitro.** To assess whether FM-scaffolds transitioned to and released spherical MCs, we loaded VS-BCP/MeO-BCP FMs with the photo-oxidizer ethyl eosin to hasten the oxidation process through exposure to white light and rapidly induce scaffold degradation in vitro (Fig. 4a). Photo-oxidation has been previously demonstrated to provide precise spatio–temporal control over transitions in PEG-bl-PPS nanostructure morphology[39]. Ethyl eosin was selected due to its hydrophobic nature (logP of 7.497)[58], which allowed partitioning within the PPS core for rapid and reproducible localized oxidation[39]. Scaffolds incorporating

10, 20, or 30% of VS-BCP and loaded with 0.75% ethyl eosin by mass were exposed to white light for varying durations of time at room temperature. Ethyl eosin loading efficiency within the FM core at 0.75% by mass was determined to be approximately 83% (Supplementary Fig. 4e, f), which aligns with previous studies encapsulating ethyl eosin within PEG-bl-PPS nanostructures[39]. CryoTEM and dynamic light scattering (DLS) were conducted on the supernatant surrounding the irradiated scaffolds, revealing monodisperse populations of spherical MCs despite varying percentages of VS-BCP (Fig. 4b, c). CryoTEM confirmed that the released nanostructures exhibited comparable size characteristics, estimated average diameters ranged from 22 to 25 nm in ImageJ, and maintained a spherical morphology regardless of the duration of time the scaffolds were exposed to oxidizing conditions (Fig. 4b). DLS analysis determined that the spherical MCs displayed number average diameters that ranged from 25 to 37 nm, and no statistically significant difference was detected between scaffolds irrespective of irradiation time or crosslinking density (Fig. 4c, Supplementary Fig. 5a). While the ImageJ and DLS determined diameters were comparable, the variation between the two measurement techniques can largely be attributed to the lack of contrast provided by the PEG corona in the cryoTEM micrographs. Due to hydration and swelling of the PEG corona when the sample is frozen in vitreous ice, there is little to no contrast with the surrounding aqueous environment[59]. As such, the ImageJ analysis of MC hydrodynamic diameter accounts for only the hydrophobic PPS core of the nanostructures. Following 24 h of irradiation, MCs released from scaffolds incorporating 10, 20, and 30% of the VS-BCP exhibited number average PDIs of 0.226, 0.131, and 0.168, respectively (Fig. 4c). These results verify that monodisperse MCs with sub-40 nm diameters were released from PEG-bl-PPS FM-scaffolds regardless of the crosslinking density and exposure time to oxidizing conditions. The size characteristics of the released nanostructures are particularly

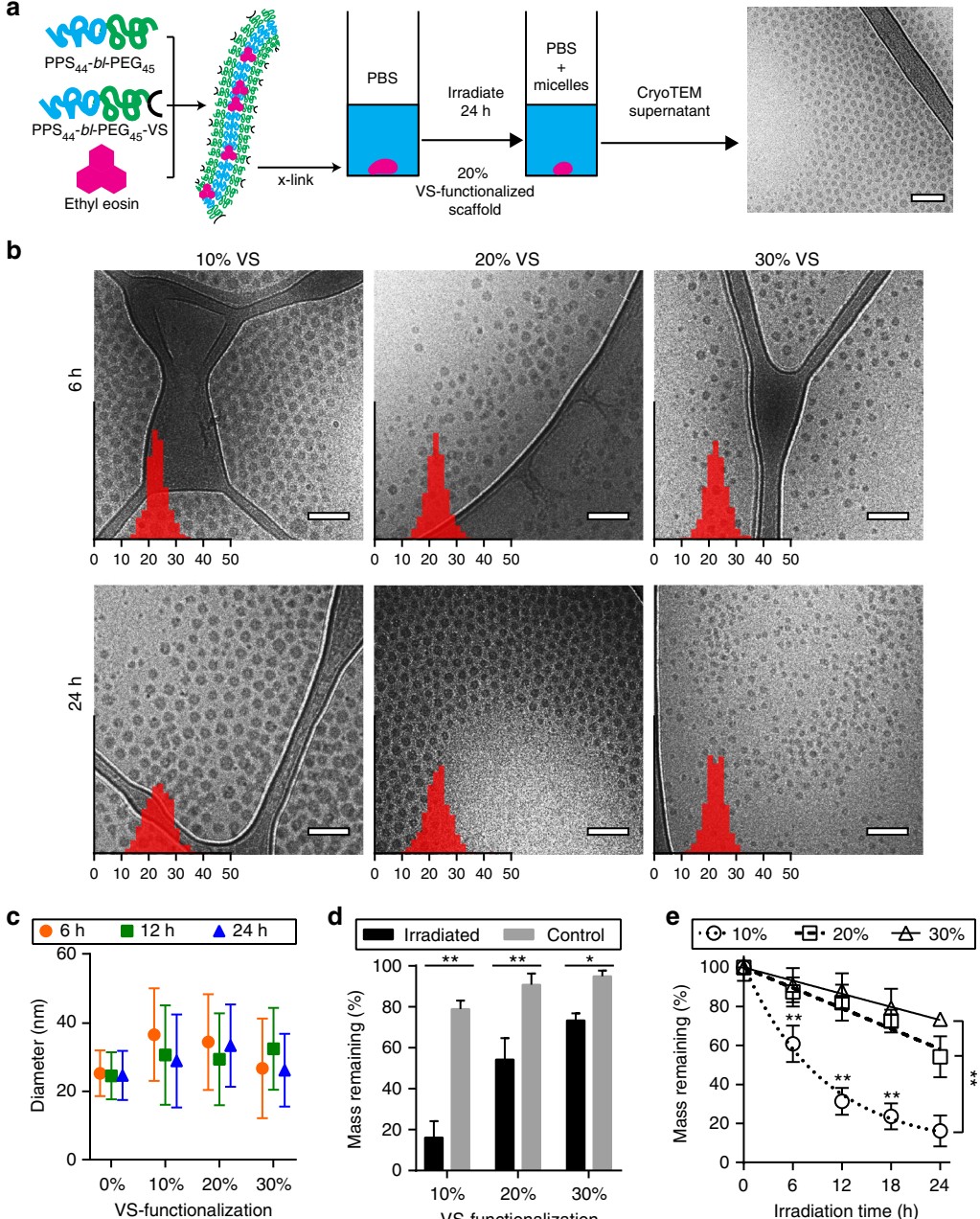

**Fig. 4** Light-induced degradation of ethyl eosin loaded PEG-bl-PPS FM-scaffolds in vitro. **a** Schematic portraying photo-oxidation of ethyl eosin-loaded FM-scaffolds, with representative cryoTEM micrograph of released PEG-bl-PPS MCs. **b** Representative cryoTEM images of micelles released into the supernatant after photodegradation of FM incorporating 10, 20, or 30% VS-BCP, irradiated for 6 or 24 h. Overlaid on images are histograms of MC diameters ($n = 500$) measured from cryoTEM images. X-axis is diameter in nanometers, y-axis is relative frequency. **c** Diameters of PEG-bl-PPS MCs released from irradiated scaffolds into supernatant, error bars represent s.e.m., $n = 3$. **d** Percent mass remaining of scaffolds irradiated for 24 h in comparison to non-irradiated controls, $n = 3$ for the control samples and $n = 7$ for the irradiated samples. **e** Modular incorporation of VS-BCP influences light induced scaffold degradation over 24 h. $n = 7$ for all samples except for $t = 0$, where $n = 3$, and scaffolds incorporating 20% VS-BCP for $t = 18$, where $n = 6$. Significance for **c** and **d** were determined with the Sidak's and Tukey's multiple comparison tests, respectively. In both cases, $*p < 0.001$, $**p < 0.0001$. Error bars represent s.d. For all cryoTEM micrographs, scale bars represent 100 nm

noteworthy as they fall within a range optimal for lymphatic transport following subcutaneous injection[60–62]. As such, MCs released from subcutaneously injected FM-scaffolds are expected to efficiently drain from the interstitial space into lymphatics, permitting delivery to lymphoid tissues such as the draining lymph nodes and spleen.

In addition to characterizing the surrounding supernatant, the remaining scaffolds were also analyzed following irradiation.

Figure 4d compares the percentage of scaffold mass remaining for irradiated scaffolds and non-irradiated controls. In the absence of irradiation, FM-scaffolds incorporating 10, 20, or 30% of the VS-BCP lost approximately 21, 9, and 5%, respectively, of their average initial mass over 24 h. But following 24 h of exposure to white light at 3.4–3.5 mW cm$^{-2}$, these percentages increased to approximately 84, 46, and 27%, respectively. Furthermore, mass differences between scaffolds containing 10% and either 20 or

30% of the VS-BCP were statistically significant at all time points tested from 6 h of irradiation onward, while those between scaffolds containing 20 and 30% of the VS-BCP were significant only after 24 h of irradiation (Fig. 4e). These trends in scaffold mass loss were corroborated by an increasing presence of monodisperse PEG-bl-PPS MCs in the supernatant (Supplementary Fig. 5b). These results demonstrated that FM modularity and crosslinking density can tailor scaffold degradation and the rate of MC release. To date, this is the only delivery system to employ the cylinder-to-sphere transition for simultaneous nanocarrier release and scaffold degradation.

**Release of intact MCs from FM-scaffold in vivo.** Having produced FM-based scaffolds and demonstrated their unique degradation mechanism in vitro, we aimed to investigate their ability to release MCs in vivo. While difficult to quantify, biologically relevant concentrations of reactive oxygen species (ROS) have been estimated to range from 50–100 µM[63], and we have previously demonstrated that $H_2O_2$ at as low as 5 µM can induce changes in PEG-bl-PPS nanocarrier morphology[6]. We therefore hypothesized that continuous exposure of FM-scaffolds to

physiologic levels of oxidation could be sufficient to induce sustained FM-to-MC transitions in vivo following subcutaneous injection in mice. Due to their stability and moderate degradation rate while releasing monodisperse MCs (Fig. 4), modular FM-scaffolds incorporating 20% VS-BCP were selected for further in vivo degradation studies. To assess the release rate, stability and cell uptake of MC from the scaffolds, we synthesized a modular system with four BCP components to permit stable in situ gelation following subcutaneous injection and multimodal imaging of MC release: MeO-BCP that comprises the majority of the FM, VS-BCP for crosslinking, DyLight 633 conjugated BCP (633-BCP) for flow cytometric analysis of MC uptake by cells, and DyLight 755 conjugated BCP (755-BCP) for near infrared whole mouse imaging of MC release (Fig. 5a). CryoTEM confirmed that incorporation of the DyLight-conjugated BCPs into the FMs did not alter FM morphology (Supplementary Fig. 6) or their ability to transition to spherical MCs (Supplementary Fig. 2d).

To verify that hydrophobic payloads and BCPs within FM-scaffolds can transfer to intact MC vehicles, as opposed to associating with lipid carrier proteins in biological fluids, we loaded the lipophilic dye DiI into the four-component modular

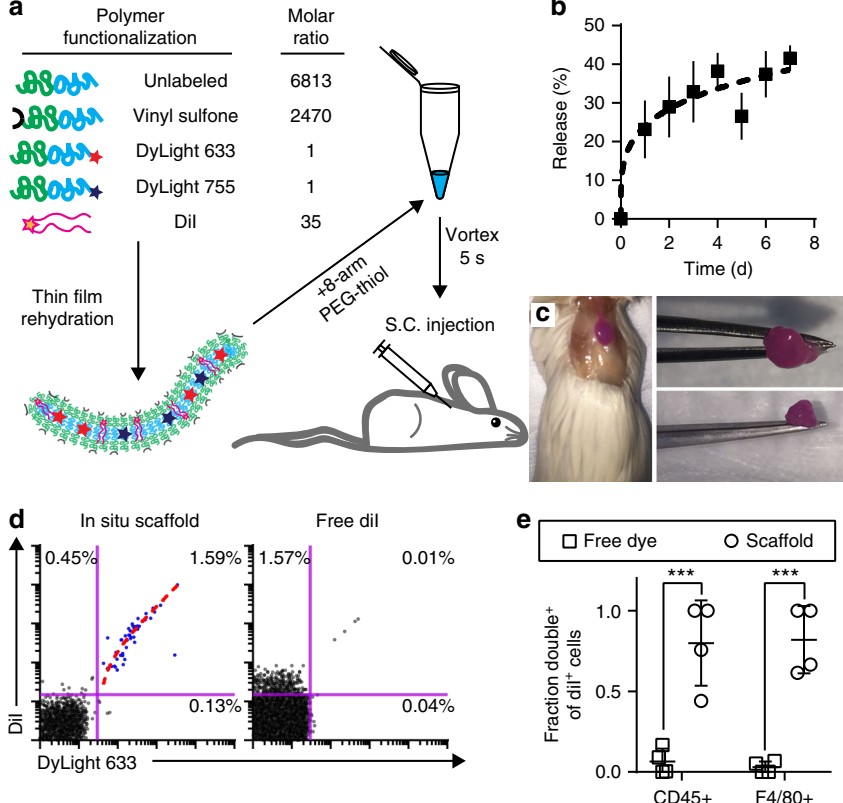

**Fig. 5** Graphical depiction of modular PEG-bl-PPS FM-scaffold preparation for in situ scaffold crosslinking and in vivo delivery. **a** Schematic of DiI-loaded PEG-bl-PPS FM modularly assembled from four separate BCPs for in situ crosslinking into FM-scaffolds and multimodal analysis following subcutaneous (S. C.) injection. **b** Intravital fluorescence imaging (IVIS) of cumulative MC (DyLight 755) release from in situ crosslinked FM-scaffolds containing 20% w/w VS-BCP over 7 days. Error bars represent SEM, n = 3. **c** Excised crosslinked scaffold 1 week after subcutaneous injection. **d** Representative flow cytometric dot plots displaying uptake of intact (double positive DiI⁺ DyLight 633⁺) released MCs by CD45⁺ cells recovered from the spleens of mice receiving injections of either DiI-loaded in situ crosslinked modular FM-scaffolds or free solubilized DiI in PBS. Percentages of events within the quadrant gates out of all events in the graph are shown. Linear fit (dotted, red) is overlaid upon the DiI⁺ DyLight 633⁺ events (blue dots), adjusted $r^2$ = 0.9773 from Pearson's correlation coefficient. Both axes are on a logarithmic scale. **e** Quantification of MC (DyLight 633) and DiI uptake by CD45⁺ or F4/80⁺ cells recovered from the spleen of mice receiving either DiI-loaded in situ crosslinked FM-scaffolds or free DiI. The y-axis represents the fraction of all DiI⁺ cells that were also DiI⁺ DyLight 633⁺ double positive. Lower fractions suggest separate uptake of solubilized DiI alone while higher fractions received intact MCs and thus both DiI and DyLight 633 simultaneously. Error bars represent s.d., n = 4 for both groups. Significance between groups assessed by Mann–Whitney U test, ***p < 0.001

FM-scaffolds and assessed cellular uptake of released MCs in vivo (Fig. 5a). Cellular colocalization of DiI fluorescence with that of 633-BCP by flow cytometry would indicate the uptake of intact MCs, while separate signals would be indicative of separate release and cellular uptake of free form and/or protein complexed 633-BCP and DiI. DiI loaded FMs were mixed with eight-arm PEG-thiol and injected subcutaneously into the scapular region of adult A/J mice for in situ crosslinking and stable adherence to surrounding tissue via VS-BCP Michael addition reactions (Fig. 5a)[64]. Intravital fluorescence imaging of the 755-BCP signal was used to monitor the release of material from the injection site over the course of 7 days (Fig. 5b, Supplementary Fig. 7a). As a control, the same volume and concentration of free form DiI in a DPBS solution was administered as a bolus injection and assessed after 24 h. Mice were subsequently sacrificed to assess MC uptake by cells within the spleen and draining lymph nodes. Recovery of the remaining hydrogel scaffold and its ex vivo manipulation verified successful in situ crosslinking and scaffold stability (Fig. 5c). For control mice, flow cytometry revealed that approximately 1.6% of extracted CD45+ splenocytes exhibited DiI fluorescence while effectively no cells were found to be double positive for DiI and 633-BCP in the spleen (Fig. 5d, e; Supplementary Fig. 7b). In contrast, cells were primarily found to be either double negative or double positive (1.6% of CD45+ splenocytes) in the FM-scaffold group, indicative of stable retention and colocalization of DiI within modular MCs containing 633-BCP during cellular uptake. The linear correlation between DiI and 633-BCP fluorescence intensity (adjusted $r^2 = 0.9773$ from Pearson's correlation coefficient) demonstrates a constant ratio of fluorescence in cells with both low and high levels of MC uptake, further confirming delivery of intact MCs containing a consistent distribution of the BCPs and loaded DiI (Fig. 5d). We calculated the percentages of DiI positive CD45+ and F4/80+ (phagocytic monocyte and macrophage populations) splenocytes that exhibited colocalized DyLight 633 fluorescence in spleen and draining lymph nodes (axillary and brachial), which revealed statistically significant differences between the FM-scaffolds and control for both cell populations (Fig. 5e, Supplementary Fig. 7b). The association of DiI fluorescence with released MCs combined with continuous loss of 755-BCP signal verify transfer of hydrophobic payloads from a scaffold depot to a nanocarrier delivery system and suggests that the cylinder-to-sphere transition can be exploited for the release of micellar delivery vehicles in a biological setting.

**Sustained FM-scaffold release of MCs for immune cell uptake.** Upon confirming that FM-scaffold degradation in vivo releases intact micellar nanocarriers, we next investigated sustained MC release and biocompatibility during bioresorption. We again employed the four-component modular FMs containing both 755-BCP and 633-BCP for a multimodal analysis via near infrared fluorescence whole mouse imaging and flow cytometry. The time course of the dissipation of the DyLight 755 signal was monitored for 28 days following subcutaneous injection of the in situ crosslinked scaffold (20%VS/MeO/633/755-BCP FMs with eight-arm PEG-thiol), non-crosslinked FMs (MeO/633/755-BCP FMs with eight-arm PEG-thiol), or bolus DyLight 755 in DPBS (Fig. 6a, Supplementary Fig. 8). The in situ crosslinked scaffold and free FMs exhibited significantly slower release from the injection site in comparison to the free form DyLight control, which dispersed over 90% of the injected material within the first five days (Fig. 6b). When comparing the in situ formed scaffolds to the non-crosslinked free form FMs, statistically significant differences in cumulative release were observed within 24 h, as the FM-scaffolds significantly reduced the amount of burst release

from approximately 65% to only 31%. The difference in percent cumulative release between the free FMs and in situ formed scaffolds remained significant at every timepoint through the first week, eventually converging after 28 days.

The lymph nodes (axillary, brachial, and inguinal), spleens, and livers were harvested following the final 1-month time point to assess cellular uptake of released MCs by flow cytometry. Tissue surrounding the injection site was recovered for histological analysis. Mice receiving in situ formed FM-scaffolds exhibited significantly greater uptake within the draining lymph nodes (axillary and brachial) than those receiving free DyLight or non-crosslinked FMs (Fig. 6c, Supplementary Fig. 9). Specifically, MHCII− dendritic cells and macrophages exhibited a discernible increase in MC fluorescence in comparison to free FM and DyLight controls. A statistically significant increase in MC fluorescence was also observed within MHCII+ dendritic cells when comparing mice receiving in situ formed FM-scaffolds in comparison to the free DyLight control. These differences in cell uptake at the 28-day time point likely reflect the differences in release rates between the crosslinked scaffolds (~0.07% mass h$^{-1}$ following the burst release) and the free form FMs (~0.02% mass h$^{-1}$ following the burst release). Uptake within the non-draining inguinal lymph nodes and liver was not statistically significant from background (Supplementary Fig. 10). Comparison of H&E and Masson's Trichrome stained tissue sections indicate only a mild increase in collagen deposition and macrophage infiltration for the mice receiving FM-scaffolds (Fig. 6d–i, Supplementary Fig. 11). Neither multi-nucleated giant cells nor signs of fibrosis were detected. These results coupled with the lack of observable symptoms associated with an injection site reaction, such as redness, swelling, blistering, infection, or weight loss (data not shown, Supplementary Fig. 12), all highlight the non-immunogenic and non-inflammatory nature of the injected FMs in free or scaffold form. This observed lack of an inflammatory response suggests that the gradual decrease in the release rate observed in Fig. 6b is not due to walling-off of the scaffold by fibrous capsule formation and may instead simply reflect the reduction in total material at the injection site over time. Employment of the cylinder-to-sphere transition thus supports sustained micellar release with non-immunogenic scaffold bioresorption, which is not possible with current alternatives for long-term nanocarrier delivery.

## Discussion

When sustained nanocarrier release is achieved via entrapment within matrices[18–22], the bulk of the construct serves no direct therapeutic role and may serve as a source for chronic inflammation. Here, we demonstrate that FM-scaffolds, composed of self-assembled PEG-bl-PPS BCP modules, can support the sustained release of monodisperse micellar nanocarriers via the cylinder-to-sphere transition with no signs of chronic inflammation-related pathology. Unlike previous depot systems, FM-scaffolds can be prepared without employing an external matrix or network to control nanocarrier retention and release. Characterization of the oxidation-dependent FM-to-MC transition was achieved via thermodynamic modeling and in vitro photo-oxidation. Photo-oxidation via a loaded ethyl eosin payload within the FM core provided a highly reproducible and temporally controllable model system. FM-to-MC-dependent degradation that would require weeks to occur under physiologic oxidative conditions was induced in a matter of hours, likely owing to the consistent ratio of ethyl eosin to PPS that was maintained by the high loading efficiency of ethyl eosin within the FM core. Following subcutaneous injection and in situ gelation in mice, our results suggest, although indirectly, that physiologic concentrations of ROS under homeostatic conditions are

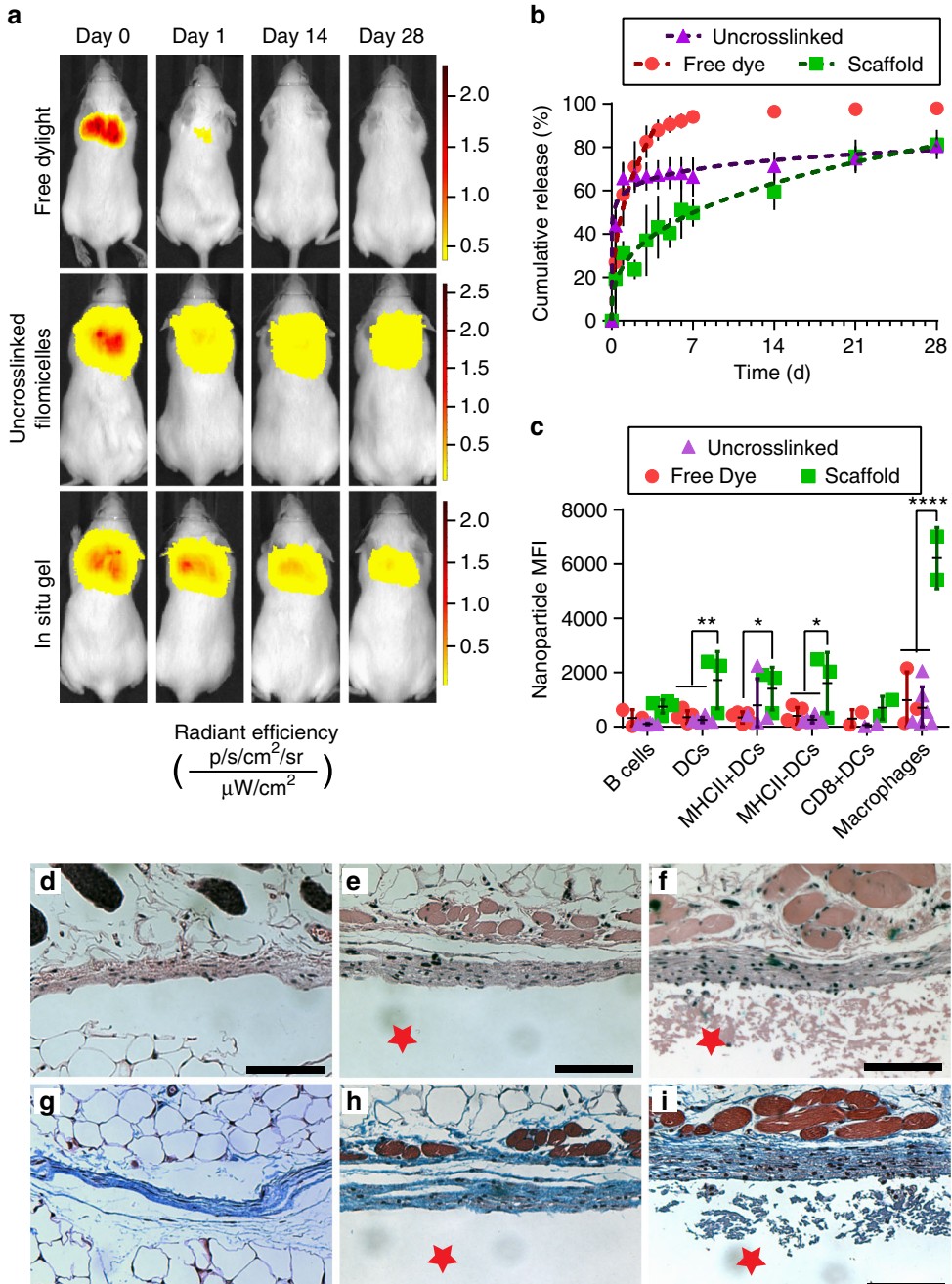

**Fig. 6** In vivo sustained release from in situ-crosslinked filomicelle scaffolds over one month. **a** Representative IVIS images of Dylight 755 injections as: free solubilized dye, conjugated to 755-BCP modules within uncrosslinked three-component modular FMs, or conjugated to 755-BCP modules within four-component FMs of in situ crosslinked scaffolds. Radiant efficiency intensity scaled per individual mouse. **b** Cumulative release curves and power law model fits of MC (Dylight 755) release from FM-scaffolds. **c** Flow cytometric analysis of MC (Dylight 633) uptake by phagocytic immune cell populations. Significance determined by Tukey multiple comparison test: *$p < 0.05$, **$p < 0.01$, ****$p < 0.0001$. For **b** and **c**, error bars represent s.d, $n = 5$ mice for free dye and $n = 4$ mice for uncrosslinked and in situ groups. Representative images of H&E staining of the interface between skin and **d** the saline Dylight 633 solution-injected control; **e** uncrosslinked FMs; or **f** the in situ crosslinked FM-scaffold, respectively. **g**, **h**, **i**, Representative images of Masson's Trichrome staining for the same groups listed above, respectively. The red star represents the scaffold or uncrosslinked FM side of the interface, for orientation purposes. All images shown are at 10× objective magnification, scale bars are 100 μm

sufficient to induce the FM-to-MC transition in vivo for sustained release of nanocarriers.

Due to the size characteristics of the released MCs, FM-scaffolds injected subcutaneously are capable of efficiently delivering encapsulated payloads to lymphoid tissues such as the draining lymph nodes and spleen. The sustained in vivo release of DyLight-conjugated MCs resulted in discernible uptake within

dendritic cells, particularly MHCII⁻ dendritic cells, and macrophages present in the draining lymph nodes. Dendritic cells and macrophages, which are professional APCs, are highly phagocytic and central players in the mononuclear phagocyte system where they are responsible for internalizing foreign material, processing antigen, and presenting antigen to T cells for activation. This ability to activate T cells coupled with their potency for cytokine

release make them central figures in dictating the body's immune response[2]. Of the professional APCs, dendritic cells and macrophages exhibit the greatest phagocytic potential and as such, are expected to effectively phagocytose nanostructures within their surrounding environment if in fact micellar structures are present. When looking at dendritic cells, MHCII expression has been found to increase during dendritic cell maturation[65] and it has been shown that mature dendritic cells may actually reduce their endocytic capacity[65]. Therefore, the statistically significant increase in nanoparticle MFI within dendritic cells, particularly MHCII⁻ dendritic cells, and macrophages described here is logical given the phagocytic capacity of these cells. Furthermore, this result highlights the potential usefulness of this construct in developing subunit vaccines and immunomodulatory treatments where the sustained delivery of encapsulated payloads to immature APCs is required. As such, these hydrogels permit highly efficient non-inflammatory bioresorption while achieving sustained nanocarrier delivery through a mechanism previously unexplored in an in vivo setting. FM-scaffolds offer an alternative to the currently employed entrapment matrices used for sustained nanoparticle delivery and have characteristics that make them highly suited for the development of immunomodulatory treatments for cancer, cardiovascular disease, and diabetes.

## Methods

**Materials**. All chemical reagents were purchased from Sigma-Aldrich St. Louis, MO, USA, unless stated otherwise. Fluorescent antibodies, Zombie Aqua fixable cell viability kit, cell staining buffer, and IC cell fixation buffer were acquired from BioLegend.

**Preparation of PEG-bl-PPS**. Syntheses began with either linear monomethoxy-poly(ethylene glycol) (mPEG, MW 2000) or linear poly(ethylene glycol) (PEG, MW 2000). mPEG (40 g, 20.0 mmol) was dried via azeotropic distillation in toluene using a Dean-Stark trap. The dried solution was cooled to room temperature while being evacuated and subsequently purged with argon (Ar). Triethylamine (16.72 mL, 120 mmol) was then added to the stirred solution, after which methanesulfonyl chloride (7.74 mL, 100 mmol) dissolved in 80 mL of toluene was added to the stirred solution drop-wise while in an ice bath. The solution was allowed to react overnight at room temperature under Ar while constantly stirring. Salt was removed via vacuum filtration of the crude product over a celite (Alfa Aesar) filter cake. Toluene was removed through rotary evaporation and the product was recovered via precipitation in ice-cold diethyl ether and dried under vacuum (percent yield = 64.1%). ¹H-NMR (400 MHz, CDCl₃): δ 3.62–3.59 (s, 180 H, PEG backbone), 3.35 (s, 3 H, O–C$\underline{H}$₃), 3.05 (s, 3 H, SO₂-C$\underline{H}$₃).

Benzyl mercaptan (53.7 μL, 0.45 mmol), dissolved in anhydrous dimethylformamide and deprotonated through the addition of sodium methoxide (0.5 M solution in methanol, 0.50 mmol), was used to initiate the living anionic ring-opening polymerization of propylene sulfide (1.50 mL, 19.94 mmol). The terminal thiolate was end-capped through the addition of mPEG-mesylate (1.57 g, 0.68 mmol) and subsequently stirred overnight. Following the removal of dimethylformamide through rotary evaporation, the product was recovered through precipitation in methanol and cooled as needed. Recovered precipitate was dried under vacuum (percent yield = 60.5%). ¹H-NMR (400 MHz, CDCl₃): δ 7.31–7.28 (d, 4H, Ar$\underline{H}$), 3.63–3.61 (s, 180H, PEG backbone), 3.36 (s, 3H, O–C$\underline{H}$₃), 2.96–2.81 (m, 88H, C$\underline{H}$₂), 2.66–2.55 (m, 44H, C$\underline{H}$), 1.38–1.32 (d, 132H, C$\underline{H}$₃).

**Preparation of VS-PEG-bl-PPS**. α-tosyl-ω-hydroxyl PEG was prepared by adapting a previously published protocol[54]. PEG (30 g, 15 mmol) was dried via azeotropic distillation in toluene using a Dean-Stark trap. Following complete removal of toluene, the remaining PEG was flushed with an Ar atmosphere and dissolved in anhydrous dichloromethane before being placed on ice. While the solution was stirring, silver (I) oxide (5.21 g, 22.5 mmol), potassium iodide (1.79 g, 10.8 mmol), and p-Toluenesulfonyl chloride (3.00 g, 15.75 mmol) were sequentially added to the solution under vigorous stirring. The solution remained on ice for two hours and was further allowed to react overnight at room temperature under Ar while constantly stirring. Salt and silver (I) oxide were removed via vacuum filtration of the crude product over a celite filter cake. Dichloromethane was removed through rotary evaporation and the product was recovered via precipitation in ice-cold diethyl ether and dried under vacuum (percent yield = 91.9%). ¹H-NMR (400 MHz, DMSO): δ 7.77–7.73 (d, 2H), 7.47–7.43 (d, 2H), 4.54–4.50 (t, 1H, O$\underline{H}$), 4.10–4.05 (t, 2H, C$\underline{H}$₂–SO₂), 3.49–3.46 (s, 180H, PEG backbone), 2.40–2.38 (s, 3H, C$\underline{H}$₃).

HO-PEG-bl-PPS was synthesized similarly to its methoxy-terminated derivative. Benzyl mercaptan (53.7 μL, 0.61 mmol), dissolved in anhydrous dimethylformamide and deprotonated through the addition of sodium methoxide

(0.5 M solution in methanol, 0.67 mmol), was used to initiate the living anionic ring-opening polymerization of propylene sulfide (2.00 mL, 26.58 mmol). The terminal thiolate was end-capped through the addition of HO-PEG-tosylate (4.08 g, 1.69 mmol) and subsequently stirred overnight. Following the removal of dimethylformamide through rotary evaporation, the product was recovered through precipitation in methanol and cooled as needed. Recovered precipitate was dried under vacuum (percent yield = 44.5%). ¹H-NMR (400 MHz, CDCl₃): δ 7.30–7.27 (d, 4H, Ar$\underline{H}$), 3.62–3.60 (s, 180H, PEG backbone), 2.95–2.80 (m, 88H, C$\underline{H}$₂), 2.65–2.54 (m, 44H, C$\underline{H}$), 1.38–1.32 (d, 132H, C$\underline{H}$₃).

The synthesis of VS-PEG-bl-PPS was adapted from a previously published protocol[66]. Lyophilized HO-PEG-bl-PPS (1.05 g, 0.02 mmol) was dissolved in anhydrous dicholoromethane and flushed with an Ar atmosphere. Sodium hydride (0.24 g, 9.90 mmol) was added to the stirring solution under Ar. Following hydrogen evolution, divinyl sulfone (0.99 mL, 9.90 mmol) was immediately added to the vigorously stirring solution. The solution was stirred under Ar for a minimum of three days. Salt was removed from the crude product through vacuum filtration over a celite filter cake. Dichloromethane was removed through rotary evaporation and the resulting product was precipitated in methanol a minimum of three times. The recovered product was dried under vacuum (percent yield = 87.3%; 85% functionalized). ¹H-NMR (400 MHz, CDCl₃): δ 7.31–7.28 (d, 4H, Ar$\underline{H}$), 6.84–6.75 (dd, 1H, SO₂–C$\underline{H}$ = CH₂), 6.40–6.33 (d, 1H, = C$\underline{H}$₂), 6.09–6.03 (d, 1$\underline{H}$, = CH₂), 3.62–3.61 (s, 180H, PEG backbone), 2.95–2.80 (m, 88H, C$\underline{H}$₂), 2.65–2.54 (m, 44H, C$\underline{H}$), 1.38–1.32 (d, 132H, C$\underline{H}$₃).

**Preparation of PEG-bl-PPS-DyLight**. PEG-thioacetate (PEG-TAA) was prepared from linear mPEG (MW 2000) as previously described[42]. PEG-TAA (1.91 g, 0.28 mmol), deprotonated through the addition of sodium methoxide (0.5 M solution in methanol, 0.33 mmol), was used to initiate the living anionic ring-opening polymerization of propylene sulfide (1.00 mL, 12.25 mmol). The terminal thiolate was end-capped through the addition of acetic acid (0.16 mL, 2.78 mmol) and subsequently stirred overnight. Following the removal of dimethylformamide through rotary evaporation, the product (mPEG-bl-PPS-SH) was recovered through precipitation in methanol and cooled as needed. Recovered precipitate was dried under vacuum (percent yield = 46.3%). ¹H-NMR (400 MHz, CDCl₃): 3.63–3.61 (s, 180H, PEG backbone), 3.36 (s, 3H, O–C$\underline{H}$₃), 2.96–2.79 (m, 88H, C$\underline{H}$₂), 2.67–2.60 (m, 44H, C$\underline{H}$), 1.38–1.30 (d, 132H, C$\underline{H}$₃).

mPEG-bl-PPS-SH (0.80 g, 0.15 mmol) was dissolved in anhydrous tetrahydrofuran and degassed with nitrogen for 30 min. DyLight 755 maleimide (ThermoFisher Scientific) (0.0005 g, 0.45 μmol) and DyLight 633 maleimide (ThermoFisher Scientific) (0.0005 g, 0.46 μmol) were added to the vigorously stirring solution. Two hours after the addition of DyLight, N-ethylmaleimide (0.18 g, 1.44 mmol) was added to the solution, which remained stirring under a nitrogen atmosphere overnight. The crude product was purified on a Sephadex-LH20 (GE Healthcare Life Sciences) gravity column and recovered in 1.0 mL fractions. Product containing fractions were pooled together and precipitated in methanol. Recovered precipitate was dried under vacuum. Column purification and precipitation were repeated to insure removal of free dye (percent yield = 83.2%).

**Preparation of FM assemblies**. FMs were generated via thin-film rehydration. PEG-bl-PPS was dissolved in ~2 mL of dichloromethane (0.5 w/v%) within 2.0 mL clear glass vials (ThermoFisher Scientific). Following the removal of dichloromethane with desiccation, the thin polymer films were hydrated with 1 mL of either Dulbecco's phosphate-buffered saline (DPBS) (ThermoFisher Scientific) or Milli-Q water and gently agitated overnight using a Stuart SB3 rotator.

**Cryogenic transmission electron microscopy (CryoTEM)**. Samples for cryoTEM were prepared by applying 3 μL of 10 mg mL⁻¹ sample on pretreated holey or lacey carbon 400 mesh TEM copper grids (Electron Microscopy Sciences). Following a 3 s blot, samples were plunge-frozen (Gatan Cryoplunge 3 freezer). Images of samples entrapped in vitreous ice were acquired using a field emission transmission electron microscope (JEOL 3200FS) operating at 300 keV with magnification ranging from 2000× to 12,000× nominal magnification. Digital Micrograph software (Gatan) was used to align the individual frames of each micrograph to compensate for stage and beam-induced drift. Any further image processing conducted on the aligned frames was completed in ImageJ.

For cryogenic electron tomography studies, FM solutions were mixed with H₂O₂ for a final H₂O₂ concentration of 0.01% by weight 30 min prior to freezing following the protocol described above. Tomographs were acquired at 12,000× nominal magnification, corresponding to a 3.4 Å pixel spacing, and a total electron dose of ~50 e⁻ Å⁻². Data was collected using SerialEM. An assymetric tilt range spanning 83° was acquired with individual images captured every 2°. The collected image series was processed using the IMOD 4.9.5 package, saved as an individual stack of images, and converted into a movie via ImageJ.

**Small angle X-ray scattering (SAXS)**. Small angle X-ray scattering (SAXS) studies were performed at the DuPont-Northwestern-Dow Collaborative Access Team (DND-CAT) beamline at Argonne National Laboratory's Advanced Photon Source (Argonne, IL, USA) with 10 keV (wavelength λ = 1.24 Å) collimated X-rays. All the samples were analyzed in the q-range (0.001–0.5 Å⁻¹), with a sample-to-detector

distance of approximately 7.5 m and an exposure time of 1 s. The diffraction patterns of silver behenate were utilized to calibrate the q-range. The momentum transfer vector q is defined as $q = 4\pi \sin\theta/\lambda$, where $\theta$ is the scattering angle. Data reduction, consisting of the removal of solvent/buffer scattering from the acquired sample scattering, was completed using PRIMUS 2.8.2 software while model fitting was completed using SasView 4.0.1 software package.

**Preparation of ethyl eosin-loaded FM-scaffolds.** FMs were generated via thin-film rehydration. Mixtures of PEG-bl-PPS and VS-PEG-bl-PPS consisting of 0, 10, 20, and 30% by mass of the VS-derived polymer were dissolved in ~2.0 mL of dichloromethane (2.5 w/v%) within 2.0 mL clear glass vials (ThermoFisher Scientific). Ethanol containing ethyl eosin was added to the vials for a final fluorophore concentration of 0.75% w/w with respect to polymer mass. Following the removal of dichloromethane and ethanol via desiccation, the thin polymer films were hydrated with 493 μL of DPBS and gently agitated for a minimum of 36 h using a Stuart SB3 rotator resulting in 10% w/v solutions of ethyl eosin-encapsulating FMs. Ethyl eosin loading efficiency was completed as described previously[39].

For scaffold formation, eight-arm PEG-thiol (Creative PEGWorks) was dissolved in DPBS to produce a 10% w/v solution. The eight-arm PEG-thiol solution was added to the 10% w/v FM solution corresponding to a 1.1:1 molar ratio of thiol:VS. The mixture was briefly vortexed before 55.0 μL aliquots were plated in 6-mm Teflon molds and cured at 37 °C for 30 min in a humidified environment to prevent evaporative loss. Following the curing procedure, scaffolds were carefully recovered and washed for 1 h in 2.0 mL reservoirs of Milli-Q water.

**Cryogenic scanning electron microscopy (CryoSEM).** Swollen scaffolds were quartered. Sections exhibiting a thickness of approximately 200 μm were further trimmed and placed in aluminum sample carriers (TECHNOTRADE International) with a 3 mm outer diameter and cavity depth of 200 μm. High-pressure freezing (HPM100, Leica) was used to preserve the internal structure of the scaffolds as seen in the wet state through vitrification. A pressure exceeding 2100 bar was applied to the samples prior to cryogenic immobilization. Vitrified samples were recovered and stored in liquid nitrogen until further processing.

Cryo-planing (UC7/FC7 Cryo-Ultramicrotome, Leica) was performed to provide a flat internal plane for imaging. While maintaining an ambient temperature of −170 °C, between 60 and 120 μm were removed from the sample surface. Planed samples were recovered from the dry nitrogen atmosphere and transferred in liquid nitrogen for freeze-etching and coating. After loading in a VCT shuttle, freshly planed samples were transferred into a freeze-etching instrument (ACE600 High Vacuum Coater, Leica) precooled to −120 °C under high vacuum. To sublimate water from the surface, the temperature was raised to −105 °C and held for 9 min. After etching the surface, the temperature was again lowered to −120 °C and the sample surface was coated with 4.0 nm of platinum and 4.0 nm of carbon to minimize charging effects. Etched and coated samples were subsequently transported via the VCT shuttle to the pre-cooled cryo-stage set at −120 °C in a field emission scanning electron microscope (S4800-II FE-SEM, Hitachi). Images were obtained at −110 °C with an accelerating voltage of 2.0 kV.

**Rheological analysis of FM-scaffolds.** Rheological analysis was conducted at 37 °C in a humidified atmosphere using a dynamic oscillatory rheometer (HR-2 DHR, TA Instruments) equipped with an 8 mm parallel plate geometry. Scaffolds were crosslinked within the rheometer gap, initially set to 0.5 mm, and allowed to cure for 30 min before being compressed to the final gap height of 0.3 mm just prior to analysis. An amplitude sweep was completed to verify that analysis was conducted within the linear viscoelastic region. Frequency dependence of the storage and loss moduli was analyzed in oscillatory mode with 0.5% applied strain ($n = 3$).

**Photoinduced oxidation of FM-scaffolds.** Visible light was used to generate singlet oxygen to induce FM oxidation and transition to spherical MCs. FM-scaffolds encapsulating ethyl eosin were incubated in 1 mL of Milli-Q water within 3.5 mL glass vials (ThermoFisher Scientific). Scaffolds were exposed to white light (Max-303 Xenon Light Source, Asahi Spectra) at an intensity of 3.4–3.5 mW cm$^{-2}$ for 6 to 24 h. Following irradiation, the 1.0 mL supernatant was removed and the remaining monolithic scaffold was lyophilized. Masses of the lyophilized scaffolds were used to assess scaffold degradation ($n = 3$ for non-irradiated control scaffolds; $n = 3$ for control scaffolds confirming average initial mass; $n = 6$ for 10% VS-BCP scaffolds irradiated for 18 h; $n = 7$ for all other treatment groups).

**Characterization of released MC.** From the recovered supernatant, 0.5 mL aliquots were diluted in 1.5 mL of Milli-Q water and analyzed for absorbance at 270 nm (SpectraMax M3, Molecular Devices) ($n = 6$ for 10% VS-BCP and 20% VS-BCP irradiated for 18 h; $n = 7$ for all other treatment groups). The size distribution of released nanostructures was obtained using a Zetasizer Nano (Malvern Instruments) equipped with a 4 mW He-Ne 633 laser ($n = 3$).

Nanostructure morphology was observed through cryoTEM, with sample preparation mimicking the protocol described above. CryoTEM was also used to confirm nanostructure size characterization provided by DLS. In brief, micrographs of MCs in the supernatant s of scaffolds irradiated for 6 and 24 h were acquired at 4000× nominal magnification. A total of 500 individual MCs, pooled from three

separate micrographs for each sample, were manually sized via ImageJ to assess micellar size distributions for each formulation at both the 6 and 24 h timepoints. Binning and histogram generation were completed in GraphPad Prism 7.03.

**Animals.** A/J female mice, 6–8 weeks old, were purchased from Jackson Laboratories. All mice were housed and maintained in the Center for Comparative Medicine at Northwestern University. All animal experimental procedures were performed according to protocols approved by the Northwestern University Institutional Animal Care and Use Committee (IACUC). A power analysis was performed to estimate the number of animals required for each experimental group. No randomization or blinding method was used to assign animals to specific groups. Statistical significance between animal groups was assessed by Mann–Whitney $U$ test or Tukey multiple comparison test.

**In vivo MC release from SC injected FM-scaffolds.** FMs were generated via thin-film rehydration. Mixtures of PEG-bl-PPS, PEG-bl-PPS-DyLight, and VS-PEG-bl-PPS consisting of 20% by mass of the VS-derived polymer were dissolved in ~2.0 mL of dichloromethane (2.5 w/v%) within 2.0 mL clear glass vials (ThermoFisher Scientific). Ethanol containing DiI (ThermoFisher Scientific) was added to the vials for a final DiI concentration, with regard to polymer mass, of 0.067% w/w. Following the removal of dichloromethane and ethanol via desiccation, the thin polymer films were hydrated with 493 μL of DPBS and gently agitated for a minimum of 36 h using a Stuart SB3 rotator resulting in 10% w/v solutions of DiI-encapsulating, DyLight-conjugated FMs.

For scaffold formation, eight-arm PEG-thiol (Creative PEGWorks) was dissolved in DPBS to produce a 10% w/v solution. A volume of the eight-arm PEG-thiol solution was added to the 10% w/v FM solution corresponding to a 1.1:1 molar ratio of thiol:VS. The mixture was briefly vortexed before 50 μL were injected subcutaneously into the scapular region of isoflurane anesthetized A/J mice for in situ crosslinking using a 28G syringe ($n = 4$). As a control, DiI solubilized in ethanol was added to DPBS with a final concentration equal to that loaded into the FMs. 50 uL of the DiI solution were injected in the same manner as the scaffold solution into separate A/J mice, 24 h before the mice were euthanized for flow cytometry ($n = 4$).

**In vivo degradation of SC injected FM-scaffolds.** FMs composed of either PEG-bl-PPS, PEG-bl-PPS-DyLight, and VS-PEG-bl-PPS (as described above) or simply PEG-bl-PPS and PEG-bl-PPS-DyLight were generated via thin-film rehydration. BCP mixtures were dissolved in ~2.0 mL of dichloromethane (2.5 w/v%) within 2.0 mL clear glass vials (ThermoFisher Scientific). Following the removal of dichloromethane via desiccation, the thin polymer films were hydrated with 493 μL of DPBS and gently agitated for a minimum of 36 h using a Stuart SB3 rotator resulting in 10% w/v solutions of DyLight-conjugated FMs.

50 μL of either a mixture of free DyLight 755 maleimide (0.03 mg) and free DyLight 633 maleimide (0.03 mg) in DPBS ($n = 5$), free FMs ($n = 4$), or VS-functionalized FMs ($n = 4$) were injected subcutaneously into the scapular region of isoflurane anesthetized A/J mice using a 28G syringe. For scaffold formation, eight-arm PEG-thiol (Creative PEGWorks) was dissolved in DPBS to produce a 10% w/v solution. A volume of the eight-arm PEG-thiol solution was added to the 10% w/v FM solution corresponding to a 1.1:1 molar ratio of thiol:VS. The mixture was briefly vortexed prior to injection and subsequent in situ crosslinking. For consistency, free FM solutions also received an equivalent volume of the eight-arm PEG-thiol solution prior to injection.

An IVIS Spectrum in vivo imaging system with a heated stage and an inhaled isoflurane manifold was used to capture intravital fluorescence images. Images were collected over the course of 28 days using filter sets of 640/745 (AF-633) and 680/800 (AF-750) with a 1.5-cm subject height. To process images, all timepoints corresponding to a single treatment group were simultaneously loaded into Living Image software. Visualization of DyLight signal was scaled per treatment rather than individual mouse. The minimum threshold value for signal visualization was increased until signal depicted on the feet and tails of all mice in the analysis was removed. Circular ROIs were applied for each mouse in the treatment group and adjusted to an area that encompassed all visible signal. Size adjusted ROIs were generated across timepoints for individual mice allowing for equivalent ROIs to be applied across all timepoints within the study. Total radiant efficiency was measured and recorded. The average background signal, recorded in an untreated A/J mouse, was used to calculate the total background radiant efficiency for each ROI. The total radiant efficiency associated with only the presence of DyLight-755 was calculated by subtracting the background radiant efficiency from the total radiant efficiency as measured in Living Image software.

**Assessment of immune cell biodistributions of released MCs.** For all the flow cytometry studies, mice were euthanized by $CO_2$ and cervical dislocation. Draining lymph nodes (brachial and axillary), non-draining lymph nodes (inguinal), spleen, and liver were collected from all mice. Liver was incubated in a collagenase solution (0.4 mg mL$^{-1}$ DNase I, 1.5 mg mL$^{-1}$ collagenase A, 5% FBS, 10 mM HEPES, in HBSS) for 45 min at 37 °C, as described previously[67], and then was processed as described for the other organs. All other organs were homogenized by mechanical disruption before being passed through a 70 μm nylon filter and washed with

RPMI. Cells were resuspended in cell staining buffer, separated into flow tubes, and were blocked with anti-CD16/CD32 and stained for cell viability using zombie aqua for 15 min. Cells were then stained with an antibody cocktail for 30 min. Cells were fixed overnight in a 1:1 solution of cell staining buffer and IC fixation buffer prior to flow cytometric analysis.

For the DiI-DyLight 633 colocalization studies, mice were euthanized 1 week after scaffold-FM injection (and 24 h after free DiI injection). The antibody cocktail used for this experiment was: PerCP/Cy5.5 anti-mouse CD45, Pacific Blue anti-mouse CD19, FITC anti-mouse CD11c, APC/Cy7 anti-mouse CD11b, PE/Cy7 anti-mouse F4/80). DiI was detected using the PE channel and DyLight 633 was detected using the APC channel.

For the one month release studies, mice were euthanized 28 days after subcutaneous injection of free DyLight, uncrosslinked FM, and FM scaffold solution. The antibody cocktails used for these studies were as follows: panel 1—Pacific Blue anti-mouse CD11c, FITC anti-mouse MHCII I-A/I-E, PerCP/Cy5.5 anti-mouse CD45, PE anti-mouse B220, PE/Cy7 anti-mouse CD8a, and APC/Cy7 anti-mouse Gr-1; panel 2—Pacific Blue anti-mouse CD11c, FITC anti-mouse F4/80, PerCP/Cy5.5 anti-mouse CD11b, PE anti-mouse CD169, PE-Cy7 anti-mouse Ly-6G, APC-Cy7 anti-mouse CD45. DyLight 633 was detected using the APC channel for both panels. Flow cytometry was performed on a BD LSRII, and data was analyzed using the CytoBank online software (Supplementary Fig. 13)[68].

**Histological analysis**. At 28 days following the subcutaneous injection of free DyLight 755 maleimide, free FMs, or VS-functionalized FMs, mice were sacrificed via carbon dioxide asphyxiation and cervical dislocation. Skin samples and surrounding injection site tissues were harvested ($n = 3$ for each treatment group) and subsequently fixed in 10% neutral buffered formalin for 96 h. Following tissue dehydration in ethanol, samples were embedded in paraffin, sectioned, and stained with haematoxylin and eosin or Masson's trichrome.

**Data availability**. All relevant data are available from authors upon request.

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

## Acknowledgements

We acknowledge staff and instrumentation support from the Structural Biology Facility at Northwestern University, the Robert H Lurie Comprehensive Cancer Center of Northwestern University and NCI CCSG P30 CA060553. The Gatan K2 direct electron detector was purchased with funds provided by the Chicago Biomedical Consortium with support from the Searle Funds at The Chicago Community Trust. SAXS experiments were performed at the DuPont-Northwestern-Dow Collaborative Access Team (DND-CAT) located at Sector 5 of the Advanced Photon Source (APS). DND-CAT is supported by Northwestern University, E.I. DuPont de Nemours & Co., and The Dow Chemical Company. This research used resources of the Advanced Photon Source, a US Department of Energy (DOE) Office of Science User Facility operated for the DOE Office of Science by Argonne National Laboratory under Contract No. DE-AC02–06CH11357. This work made use of the EPIC facility of Northwestern University's NUANCE Center, which has received support from the Soft and Hybrid Nanotechnology Experimental (SHyNE) Resource (NSF ECCS-1542205); the MRSEC program (NSF DMR-1121262) at the Materials Research Center; the International Institute for Nanotechnology (IIN); the Keck Foundation; and the State of Illinois, through the IIN. This work made use of the IMSERC at Northwestern University, which has received support from the NSF (CHE-1048773); Soft and Hybrid Nanotechnology Experimental (SHyNE) Resource (NSF NNCI-1542205); the State of Illinois and International Institute for Nanotechnology (IIN). Histology services were provided by the Northwestern University Mouse Histology and Phenotyping Laboratory which is supported by NCI P30-CA060553 awarded to the Robert H Lurie Comprehensive Cancer Center. This work was supported by the Northwestern University—Flow Cytometry Core Facility supported by Cancer Center Support Grant (NCI CA060553). Imaging work was performed at the Northwestern University Center for Advanced Molecular Imaging generously supported by NCI CCSG P30 CA060553 awarded to the Robert H Lurie Comprehensive Cancer Center. The authors acknowledge Dr. Reiner Bleher (NUANCE Center-EPIC, NU) and Jonathan Remis (Structural Biology Facility, NU) for their contribution to cryoSEM and cryoTEM image acquisition, respectively. This research was supported by the National Science Foundation CAREER award CBET-1453576, the National Institutes of Health Director's New Innovator Award no. 1DP2HL132390-01 and the 2014 McCormick Catalyst Award. N.B.K. was supported in part by the Northwestern University Graduate School Cluster in Biotechnology, Systems, and Synthetic Biology, which is affiliated with the Biotechnology Training Program. T.S. and E.F. are grateful to the National Science Foundation CAREER award DMR-1564950 for providing partial financial support. H.K.K was funded by the Center for Hierarchical Materials Design (ChiMaD).

## Author contributions

N.B.K. and E.A.S. contributed to the conception and study design. N.B.K. synthesized and characterized materials. N.B.K. and S.B. contributed to SAXS characterization of nanostructures and corresponding analysis. N.B.K. completed the in vitro experiments. H.-K.K. and K.R.S. contributed to the conception, design, and data acquisition of the applied thermodynamic model. N.B.K., E.F. and T.S. contributed to the TEM imaging. N.B.K. and S.D.A. conducted the in vivo experiments. N.B.K., S.D.A., and E.A.S. contributed to the statistical analysis. N.B.K., S.D.A., H.-K.K., and E.A.S. contributed to writing the manuscript.

## Additional information

**Competing interests:** The authors declare no competing financial interests.

