## [Peer Review File · Nature Communications]

Reviewers' comments:

Reviewer #1 (Remarks to the Author):

In this work the authors describe an injectable scaffold delivery system that exploits a transition from rod-like to spherical micelles to both deliver and degrade the scaffold. This is important because as opposed to other systems where the scaffold serves as a passive matrix to incorporate the delivery system; here the scaffold itself, in the form of crosslinked rod-like block-co-polymer micelles, is responsible for release and bio-resorption. The work is important and well conducted but its novelty hinges on the rod-sphere micellar transition (in crosslinked micelles and in-vivo). My concerns with the paper relate to this mechanism or the way it is described. It is my opinion that if the authors appropriately address the following concerns, the work is suitable to be presented at Nature Communications. My comments are enumerated below:

1. The first part of the study states that the block-co-polymer (BCP) system used self-assembles into rod-like micelles (FMs) that transition to "more thermodynamically stable spherical micelles (MCs)". The transition is compared to what is observed in other BCP systems. [Figure 1]. I find the description of transition to "more thermodynamically stable MCs" due to surface tension incomplete. Depending on the BCP system, one can in principle attain FMs that are thermodynamically stable due to molecular packing reasons. Surface tension is just one factor dictating the morphology of the micelles.
2. The authors present Cryo-EM results in Fig. 1 as evidence that MCs bud off the FM ends, and the process is compared to the work of He et al (ACS Macro 2014). In that work it is shown that Rayleigh instabilities lead to significant FM pearling followed by MC formation. It is not just budding at the end. I find that the data presented is not convincing enough of fluctuations present in the FM morphologies. Also, budding at the end is not very clear. How can it be excluded that what we see are FM micelles head on? The only way to resolve this issue is to do Cryo-EM tomography. Alternatively, the authors may collect more data where undulations of FMs and budding is more clearly depicted.
3. One important assumption is that the FM-MC transitions occur even when FMs are crosslinked to form the scaffold. If the driving force for FM-MC transition relies just on surface tension, it is hard to imagine this would be the case. Undulation instabilities require a highly dynamic system and crosslinked FMs do not appear very dynamic.
4. There is no Cryo-EM (or other structural tool) clearly demonstrating the FM-MC transition at the crosslinked state. Instead the authors load the micelles with a photo-oxidizer and monitor the release. [Figure 3]. The method is compared to the work of Hubble et al. (ACS Nano 2012). What is the loading efficiency of the photo-oxidizer? How do we know that its exact location is at the FM core?
5. The work of Ref8. is different because there photo-oxidizers are located in vesicles that get ruptured into a polydisperse system of smaller vesicles and micelles upon light exposure. This mechanism of membrane disruptions can not be compared with any depth to a spontaneous FM-MC transition based on surface tension minimization.
6. The supernatant is evaluated and it appears to be composed of a remarkably monodisperse collection of micelles (not a polydispersity mix as in ref.8 caused by disruption). Why should the system form such a monodisperse collection of MCs? Is it because it is a result of a Rayleigh instability? A discussion of this is lacking. Cryo-EM images of the supernatant control systems with no light irradiation need to be presented.
7. It is concerning that the structures assigned to micelles in the Cryo-EM images seem to measure around 10-15 nm not matching at all with the DLS results (25-37 nm).
8. When discussing DyLight-conjugated BCPs and Cryo data, it is supposed to be supplementary Fig. 4 not 5.
9. Does DyLight-conjugated BCPs affect the crosslinking behavior? If the DyLight-conjugated BCPs do not affect FM morphology, why don't we see MCs budding off FMs micelles in the Cryo-EM images as it is postulated is happening for these systems?
10. I think it is an overstatement to say that an in-vivo FM-MC transition was observed for the first

time. This is only inferred indirectly from release data. Release could have happened due to all sorts of things like degradation, passive release etc.

11. Are the MC sizes produced suitable for a delivery application other than the spleen, liver?

Reviewer #2 (Remarks to the Author):

Review attached

Reviewer #3 (Remarks to the Author):

This paper by Karabin et al. reports on a novel strategy for sustained drug delivery, based on the use of a thermodynamic/kinetic transition of block copolymers between a filomicelle/nanofiber morphology and spherical micelles as a means to create a reservoir of polymer that steadily releases spherical micelles over extended time periods. The transition is mediated by oxidation of poly(propylene sulfide) block copolymers, a system previously studied in detail by this group. The authors demonstrate both triggered and spontaneous release of micelles from nanofiber matrices, via photo-oxidation or spontaneous oxidation in vivo. Rates of mass loss from the fibers are shown to be tunable by crosslinks introduced between the fibers, and data is presented suggesting successful delivery of a dye surrogate of a hydrophobic drug to cells of the lymph nodes and spleen by released micelles. This is an exciting, elegant concept and clever exploitation of known structural transitions between block copolymer morphologies. The experiments are well executed and the data is clearly presented. Although the authors show only release of a "model drug", the concept is compelling.

Some minor issues that should be addressed:

1. Supplementary Figure 4 is incorrectly called out in the text as Suppl Fig. 5.
2. The free dye control case reported in Supplementary Fig. 5 is confusing: If there is not Alexa633-labeled block copolymer present in the free dye case, how can there be a double+ population of cells? If the gating were rigorously excluding autofluorescence, this population should be by definition zero. This analysis should be revisited/clarified.
3. It would be useful for the authors to report on the % of nanoparticle+ cells to accompany Fig. 5c, or show raw histograms of micelle signal in the different cell populations in supplemental, to give the reader a better sense of how much material is still present at the late time point shown.
4. Why does dye release plateau for both the uncrosslinked and crosslinked scaffold groups at ~75% released, rather than steadily continuing on toward 100% release at late times? Could this reflect that ~25% is being phagocytosed by sessile macrophages at the injection site? Or a portion walled off by the host response and unable to exit the tissue? This is an important point that should be clarified.

Response to Reviewers' comments:

Referee 1:

1. The first part of the study states that the block-co-polymer (BCP) system used self-assembles into rod-like micelles (FMs) that transition to “more thermodynamically stable spherical micelles (MCs)”. The transition is compared to what is observed in other BCP systems. [Figure 1]. I find the description of transition to “more thermodynamically stable MCs” due to surface tension incomplete. Depending on the BCP system, one can in principle attain FMs that are thermodynamically stable due to molecular packing reasons. Surface tension is just one factor dictating the morphology of the micelles.

Reviewer 1 makes an excellent point, and we did not intend to suggest that surface tension was the only means to induce FM-to-MC transitions nor that stable FMs could not be obtained. We removed the stated sentence and rewrote the introduction for clarity. Of note, we performed additional work in collaboration with Dr. Kenneth Shull at Northwestern University to further verify that surface tension could have influence over the morphology of our nanostructures. Through the use of a thermodynamic model and the acquisition of interfacial tension measurements for both PEG-*bl*-PPS and its oxidized derivative, we demonstrate how the PEG-*bl*-PPS FM-to-MC transition can be explained through a reduction in interfacial energy that occurs as a result of the oxidation of the sulfide group within the propylene sulfide monomer. This data is presented in a new Figure 2.

2. The authors present Cryo-EM results in Fig. 1 as evidence that MCs bud off the FM ends, and the process is compared to the work of He et al (ACS Macro 2014). In that work it is shown that Rayleigh instabilities lead to significant FM pearling followed by MC formation. It is not just budding at the end. I find that the data presented is not convincing enough of fluctuations present in the FM morphologies. Also, budding at the end is not very clear. How can it be excluded that what we see are FM micelles head on? The only way to resolve this issue is to do Cryo-EM tomography. Alternatively, the authors may collect more data where undulations of FMs and budding is more clearly depicted.

We thank the reviewer for the comments and useful suggestions. We referenced the work by He *et al* to highlight potential mechanisms for the cylinder-to-sphere transition. We do not in fact believe that our system undergoes Rayleigh-like budding. As mentioned above, thermodynamic modeling performed with the Shull lab suggests that oxidation dependent changes in surface tension are instead likely inducing the observed budding. We have adjusted the text for clarity.

We understand the concerns raised about the clarity of the cryoTEM micrographs. To supplement the cryoTEM micrographs included in the original submission, we provided additional images of budding in revised Supplementary Figure 1. With these additional images, we can see what appears to be elongation occurring where the cylindrical FM body meets the bulbous FM endcap. In some instances, it is possible to see what appears to be a lone undulation adjacent to this elongated region on the FM. That being said, this undulation is most likely the soon-to-be bulbous endcap rather than a “pearling” state associated with Rayleigh instabilities. As such, we believe these cryoTEM micrographs suggest that the PEG-*bl*-PPS FM-to-MC transition occurs through a budding process restrained to the FM ends.

Page 6: “We further employed cryoTEM to capture morphologic transitions at the high curvature ends of FMs assembled from MeO-BCPs (**Fig. 1d,e, Supplementary Fig. 1**).

Additionally, the suggestion to pursue cryo-EM tomography by the reviewer was a fantastic idea. Please see the included Supplementary Video 1. The video demonstrates that the MCs observed are clearly spherical MCs and not simply FMs oriented perpendicularly to the surface the of the grid. We have edited the text extensively. Key changes to the text are listed below:

Page 6: Three-dimensional cryoTEM tomography verified that the depicted MCs were not the result of FMs oriented perpendicularly to the sample grid but were in fact a separate morphology (**Supplementary Video 1**).

Page 20: For cryogenic electron tomography studies, FM solutions were mixed with H₂O₂ for a final H₂O₂ concentration of 0.01% by weight 30 minutes prior to freezing following the protocol described above. Tomographs were acquired at 12,000 x nominal magnification, corresponding to a 3.4 Å pixel spacing, and a total electron dose of ~50 e⁻ Å⁻². Data was collected using SerialEM. An asymmetric tilt range spanning 83° was acquired with individual images captured every 2°. The collected image series was processed using the IMOD 4.9.5 package, saved as an individual stack of images, and converted into video via ImageJ.“

3. One important assumption in that the FM-MC transitions occur even when FMs are crosslinked to form the scaffold. If the driving force for FM-MC transition relies just on surface tension, it is hard to imagine this would be the case. Undulation instabilities require a highly dynamic system and crosslinked FMs do not appear very dynamic.

Reviewer 1 raises good points in that we do not directly image the FM-to-MC transition of crosslinked scaffolds and that Rayleigh instabilities would require a highly dynamic system. Our response here addresses Question 3 above as well as parts of Question 4 below. First, we would like to again note that we do not believe that Rayleigh instability is the mechanism of our FM-to-MC transitions. We were originally intending to save this additional modeling work for a subsequent publication, but have instead provided the data here in Figure 2 and a new results section entitled “Thermodynamic modelling of FM-to-MC transition” to better describe our system and address multiple questions about the mechanism of MC release. Our collaborator Dr. Shull is an expert in modeling the thermodynamics and resulting geometries of self-assembled gels formed from block copolymers (BCPs). As described in the new text and figure, thermodynamic modeling results demonstrate that PEG-*bl*-PPS oxidation is sufficient to induce dynamic changes in surface tension for induction of FM-to-MC transitions.

Second, we attempted to use cryoTEM to image frozen scaffolds, but due to the thickness and density of crosslinked hydrogel samples, we were unable to acquire micrographs with sufficient detail and contrast to verify this transition. But, our initial submission clearly demonstrates that: 1) oxidation of PEG-*bl*-PPS results in a significant mass loss in comparison to unoxidized controls and 2) this mass loss is coupled with an increase in PEG-*bl*-PPS MCs within the surrounding supernatant, confirmed through a combination of absorbance measurements, dynamic light scattering, and cryoTEM. This data suggests that, while we are unable to capture this transition in the crosslinked form via cryoTEM, this FM-to-MC transition is the result of oxidation and continues to occur in the presence of stabilizing crosslinks. It should be noted that only between

10% to 30% (20% for most samples) of available BCPs can participate in crosslinks in the presented hydrogels, and thus the majority of FM nanostructures even in crosslinked form remain unmodified by the PEG crosslinker and are available to undergo transitions like the native uncrosslinked FMs.

To further verify that the scaffold underwent FM-to-MC transitions, we repeated the photodegradation experiment shown in the original Figure 3 to increase the replicates to at least 6 for each sample. We additionally performed cryoTEM on the supernatant from multiple scaffolds following 6 and 24 h of oxidation-induced degradation to assess the structure of released degradation products. CryoTEM strongly supported our previous results, showing only monodisperse micelles being generated. These images were added to the paper and can be observed as a new Figure 4. Of note, the 24 h time point resulted in a mass decrease of over 80% for the 10% VS-BCP scaffolds, and cryoTEM and DLS both demonstrate the presence of only monodisperse micelles in the supernatant. These scaffolds were thoroughly washed to ensure no transfer of any residual free form MCs, so all micelles found in the supernatant were generated by the degraded FM-scaffold.

4. There is no Cryo-EM (or other structural tool) clearly demonstrating the FM-MC transition at the crosslinked state. Instead the authors load the micelles with a photo-oxidizer and monitor the release. [Figure 3]. The method is compared to the work of Hubble et al. (ACS Nano 2012). What is the loading efficiency of the photo-oxidizer? How do we know that its exact location is at the FM core?

For the first part of this question concerning the FM-MC transition in the crosslinked state, please see our response to Question 3 above.

Reviewer 1 brings up a great point that should have been included in our initial submission. The loading efficiency of 0.75% w/w ethyl eosin in PEG-*bl*-PPS FMs is approximately 83%. This aligns well with our previous work (Vasdekis & Scott *et al.* ACS Nano 2012) that quantified an ethyl eosin loading efficiency of 84-93% within PEG-*bl*-PPS polymersomes. As described in their previous work, loading efficiency was quantified following nanostructure and dye separation on a Sepharose 6B column. Supplementary Fig. 3e,f were added to the manuscript to show this data.

We have previously demonstrated (Vasdekis & Scott *et al.*; Scott *et al.* Biomaterials 2012; and Allen *et al.* J. Control Release 2017) that the loading of small molecules with hydrophobic character, such as ethyl eosin, within PEG-*bl*-PPS nanostructures occurs within the hydrophobic volume off the PPS core or membrane. These references have been included in the manuscript. Vasdekis & Scott *et al.* thoroughly characterizes the ethyl eosin-loaded PEG-*bl*-PPS assemblies, and our results were consistent with this previous study. We have furthermore assessed the loading of payloads into PEG-*bl*-PPS nanocarrier hydrophobic cores based on logP values. In our recent publication (Allen *et al.*, J. Control. Release 2017) the logP value for ethyl eosin was pulled from the ZINC15 database (zinc15.docking.org) and determined to be ~7.5, suggesting favorable partitioning into the hydrophobic region of the assemblies, which was supported by the data in that publication.

Although the size exclusion column purification should remove the vast majority of ethyl eosin not loaded within the FMs, any ethyl eosin not encapsulated within the PPS core could still play a

secondary role in oxidation given that singlet oxygen has been calculated to diffuse a few hundred nanometers in physiologic conditions (Skovsen *et al*, J. Phys. Chem. B 2005).

The following text was updated for clarity:

Page 9: “Ethyl eosin was selected due to its hydrophobic nature (logP of 7.497)⁵⁷, which allowed partitioning within the PPS core for rapid and reproducible localized oxidation³⁹.”

Page 9: “ Ethyl eosin loading efficiency within the FM core at 0.75% by mass was determined to be approximately 83% (**Supplementary Fig. 3e,f**), which aligns with previous studies encapsulating ethyl eosin within PEG-*bl*-PPS nanostructures³⁹.”

5. The work of Ref8. is different because there photo-oxidizers are located in vesicles that get ruptured into a polydisperse system of smaller vesicles and micelles upon light exposure. This mechanism of membrane disruptions can not be compared with any depth to a spontaneous FM-MC transition based on surface tension minimization.

We thank the reviewer for this comment. First, while it is fair to point out that the transition from vesicular structure to micelle is not a direct comparison to the FM-to-MC transition utilized in this work, the application of photooxidation within the two works is nonetheless the same. The reference to the work by Vasdekis & Scott demonstrates that: 1) ethyl eosin has been previously used as a photooxidizer and 2) the generation of singlet oxidation by ethyl eosin is significant enough to oxidize PEG-*bl*-PPS. It should be noted that a thermodynamic explanation for the oxidation induced transition from vesicles to MC was not described in the Vasdekis & Scott paper. We have addressed this question in the updated version of our manuscript (Figure 2) and propose a more thorough explanation for these observations.

Second, in the *in vitro* experiment discussed in Figure 4, we are not relying on spontaneous FM-to-MC transitions but are instead generating reactive oxygen species (ROS) (like singlet oxygen) to induce this morphologic change. It should be noted this experiment was not included to sell this platform as an externally controlled stimuli responsive system (although we do plan to eventually pursue this route in the future), but instead, we wished to highlight how this platform can transition into spherical MCs under oxidative conditions and that we exhibit control over the relative release rate of said MCs by adjusting FM functionalization. The use of photooxidation in this instance is particularly useful because it allows us to 1) induce the FM-to-MC transition more quickly in a shorter and controlled timeframe and 2) to consistently generate ROS *in situ* in close proximity to the FM PPS core by specifying a consistent concentration of loaded ethyl eosin.

The following text was added for clarity:

Page 9: “Ethyl eosin was selected due to its hydrophobic nature (logP of 7.497)⁵⁸, which provided partitioning within and close proximity to the PPS core for rapid and reproducible oxidation³⁹”

Page 15: “Characterization of the oxidation-dependent FM-to-MC transition was achieved via thermodynamic modeling and *in vitro* photo-oxidation. Photo-oxidation via a loaded ethyl eosin payload within the FM core provided a highly reproducible and temporally controllable model system. FM-to-MC dependent degradation that would require weeks to occur under physiologic oxidative conditions was induced in a matter of hours, likely owing to the close proximity of loaded

ethyl eosin to the PPS blocks within the FMs and the consistent ratio of ethyl eosin to PPS that was maintained by the high loading efficiency. “

6. The supernatant is evaluated and it appears to be composed of a remarkably monodisperse collection of micelles (not a polydispersity mix as in ref.8 caused by disruption). Why should the system form such a monodisperse collection of MCs? Is it because it is a result of a Rayleigh instability? A discussion of this is lacking. Cryo-EM images of the supernatant control systems with no light irradiation need to be presented.

The reviewer’s comment is much appreciated, as we also find the monodispersity of the micelles to be quite remarkable. As previously described above, the newly included thermodynamic model indicates that the PEG-*b*-PPS FM-to-MC transition occurs as a consequence of a drop in interfacial energy following sulfide oxidation. Additional cryoTEM micrographs visualize that this transition occurs through end budding and not through Rayleigh instability. And yes, we do believe that the formation of micelles from the budding process may be responsible for this observed monodispersity. After each micelle buds, the end of the FMs will be consistently exposed to the same interfacial conditions as the previous budding process. Further modeling and investigation into this process will be necessary, and we believe that this should be included in a separate paper focused on this topic.

We believe the mass loss data provided in Figure 4d highlights that even without an oxidizing stimulus, mass loss may occur, albeit at a much slower rate. This loss is most likely attributable to FMs undergoing spontaneous transitions. We have already provided cryoTEM images of MCs formed from spontaneous FM-to-MC transitions in Figures 1d and 1e, which both verify the monodispersity of sequentially released MC. Considering the extensive cryoTEM completed for both the initial submission and revised submission and the low levels of MCs present in the supernatant without oxidation, we decided to instead focus our resources on further characterizing the MC released by the various scaffolds at different levels of crosslinking. This new data is presented in Fig. 4b.

7. It is concerning that structures assigned to micelles in the Cryo-EM images seem to measure around 10-15 nm not matching at all with the DLS results (25-37 nm).

Reviewer 1 raises an excellent point and one that deserves an explanation within the manuscript. Hydration of the PEG corona limits its visibility during cryoTEM as there is no visible contrast between the hydrated layer and the surrounding vitreous ice. As such, the dark grey structures observed in the provided cryoTEM images represent only the PPS core of the FMs and MCs. A discrepancy between the cryoTEM measured diameter and DLS calculated diameter is expected. Similar discrepancies have been reported previously (Pinol *et al*, *Macromolecules* 2007). Adjustments to the text were as follows:

Page 10: “While the ImageJ and DLS determined diameters were comparable, the variation between the two measurement techniques can largely be attributed to the lack of contrast provided by the PEG corona in the cryoTEM micrographs. Due to hydration and swelling of the PEG corona when the sample is frozen in vitreous ice, there is little to no contrast with the surrounding aqueous environment⁵⁸. As such, the ImageJ analysis of MC hydrodynamic diameter accounts for only the hydrophobic PPS core of the nanostructures.”

Furthermore, we were able to obtain beam time at Argonne National Laboratory and performed SAXS analysis on the FMs. This work resulted in an estimated FM core thickness (also doesn't include the PEG length) of ~10 nm as well. A new Figure 1c was included for the SAXS analysis.

Additionally, we have provided new micrographs of the surrounding supernatant for FM-scaffolds irradiated for 6 and 24 hours (Figure 4a,b). These added micrographs are at a higher magnification and include size distributions recorded through manual sizing using ImageJ software.

8. When discussing DyLight-conjugated BCPs and Cryo data, it is supposed to be supplementary Fig. 4 not 5.

We thank the reviewer for catching this error. We have corrected the labeling within the manuscript.

9. Does DyLight-conjugated BCPs affect the crosslinking behavior? If the DyLight-conjugated BCPs do not affect FM morphology, with don't we see MCs budding off FMs micelles in the Cryo-EM images as it is postulated is happening for these systems?

While cryoTEM micrographs are representative of the samples in solution, there exists variability across the grid. Upon closer inspection, budding is in fact visible from the ends of some DyLight-conjugated FMs in Supplementary Figure 5. Regardless, lower levels of budding depicted in the images of DyLight conjugated FMs in Supplementary Figure 5 is not indicative of a lack of potential for budding upon oxidation. The colocalization experiment depicted in Figure 6 suggests that the DyLight-conjugated BCPs do not inhibit this budding process. This can be concluded from Figure 5e, which suggests the release of intact nanostructures carrying the hydrophobic DiI. To further allay such concerns, we have included a micrograph of budding occurring in FMs incorporating DyLight-conjugated BCPs in Supplementary Figure 1d.

10. I think it is an overstatement to say that an in-vivo FM-MC transition was observed for the first time. This is only inferred indirectly from release data. Release could have happened due to all sorts of things like degradation, passive release etc.

The reviewer is correct in that we do not directly show the FM-MC transition *in vivo*, which would be an incredibly difficult task. As stated by the reviewer, we were forced to infer indirectly that stable MCs were released based on colocalization of intracellular fluorescence as detected by flow cytometry. Such colocalization of BCP fluorescence with DiI fluorescence strongly suggests that passive release of the FM-loaded DiI did not occur. To avoid the risk that our choice of words could be considered an overstatement, the manuscript has been edited as follows:

Page 13: "The association of DiI fluorescence with released MCs combined with continuous loss of 755-BCP signal verify transfer of hydrophobic payloads from a scaffold depot to a nanocarrier delivery system and suggests that the cylinder-to-sphere transition can be exploited for the release of micellar delivery vehicles in a biological setting."

The following text was added to the discussion for clarity:

Page 16: "Following subcutaneous injection and *in situ* gelation in mice, our results suggest, although indirectly, that physiologic concentrations of ROS under homeostatic conditions are sufficient to induce the FM-to-MC transition in vivo for sustained release of nanocarriers."

11. Are the MC sizes produced suitable for a delivery application other than the spleen, liver?

The point raised by Reviewer 1 is important to the significance of the presented work, and highlights our failure to fully reinforce this point in the initial submission. The works by Reddy *et al* and Oussoren *et al* demonstrate that nanoparticle access to the lymphatics is size dependent. Specifically, Reddy *et al* observed that nanoparticles with diameters ranging from 20 – 45 nm were effective at accessing the lymphatics from interstitial tissue. The released PEG-*bl*-PPS MCs exhibit diameters that fall within this range, making them useful delivery vehicles for targeting immune cell populations within the lymph nodes and spleen. This characteristic of the released MCs is highlighted in Figure 6c. To further clarify this, the figure caption has been adjusted as follows:

Fig. 6: “Flow cytometric analysis of MC (Dylight 633) uptake by phagocytic immune cell populations in the draining lymph node.”

Additional text was added to the manuscript to further highlight the importance of MC size:

Page 11: “The size characteristics of the released nanostructures are particularly noteworthy as they fall within a range optimal for lymphatic transport following subcutaneous injection⁵⁹⁻⁶¹. As such, MCs released from subcutaneously injected FM-scaffolds are expected to efficiently drain from the interstitial space into lymphatics, permitting delivery to lymphoid tissues such as the draining lymph nodes and spleen.”

Referee 2:

1. Please provide detailed chemical reaction mechanisms and schematics for each synthesis step and each compound prepared in this manuscript. The schematics shown in Supplementary Figure 1 are not sufficient. Please combine all chemical reactions and individual reaction/synthesis steps shown in Figure 1 and Supplementary Figure 1 into one single Figure. Please double-check reaction conditions, reagents, and solvents. For example, did authors use DCM or toluene for synthesis of mPEG-mesylate (Supplementary Figure 1)?

We appreciate the suggestion provided by Reviewer 2. Initially we included only previously unpublished materials in our schematics. Revised Supplementary Schema 1 depicts all the syntheses completed within the manuscript and lists the primary reagents, solvent, and temperature utilized for each synthesis.

2. Please provide ¹H-NMR and MS analyses for each compound synthesized in this study. What are the corresponding reaction yields for each compound?

NMR spectra and GPC chromatograms have been included for the previously unpublished materials, HO-PEG₄₅-*bl*-PPS₄₄-Bn and VS-PEG₄₅-*bl*-PPS₄₄-Bn, synthesized for this publication. Spectra and chromatograms for the remaining materials, which have been previously published, have been omitted. Yields for all the syntheses described in the Methods section have now been incorporated. The following yields are listed in their corresponding section (please note that this

table is for the convenience of the reviewer and was not included as a figure in the manuscript since this data is already included in the updated Methods section):

Material	Yield (%)
mPEG ₄₅ -OMs	64.1
MeO-PEG ₄₅ - bl -PPS ₄₄ -Bn	60.5
HO-PEG ₄₅ -OTs	91.9
HO-PEG ₄₅ - bl -PPS ₄₄ -Bn	44.5
VS-PEG ₄₅ - bl -PPS ₄₄ -Bn	87.3
MeO-PEG ₄₅ - bl -PPS ₄₄ -SH	46.3
MeO-PEG ₄₅ - bl -PPS ₄₄ - DyLight	83.2

Mass spectrometry was not completed for any materials completed in this publication.

3. Please move the cartoons of PPS₄₄-*bl*-PEG₄₅ and PPS₄₄-*bl*-PEG₄₅-VS polymers into a separate panel of Figure 1. Make the filomicelle larger so that the details and features of self-assembled polymers are more evident.

This was a great suggestion as the initial FM depiction lacked sufficient detail. We have modified Figure 1 so that the FM cartoon is now clearly composed of individual PEG-*bl*-PPS molecules.

4. It would strengthen the manuscript, if authors would provide theoretical calculations and mathematical models to explain the observed cylinder-to-sphere transitions from a quantitative thermodynamic perspective.

We are in agreement with the reviewer and were planning to submit a separate subsequent publication on this topic. In order to address this question as well as several others, we decided to include this data in the present manuscript as a new Figure 2. Through a collaboration with Dr. Kenneth Shull at Northwestern University, we were able to include a model with our revisions. Inclusion of this model allowed us to show how the PEG-*bl*-PPS FM-to-MC transition can be explained through a reduction in interfacial energy that occurs following the oxidation of the sulfide group within the propylene sulfide monomer. While this model does not allow us to quantitatively estimate the drop in interfacial tension required to induce this transition, it should be noted that this model does provide a qualitative confirmation of interfacial tension-driven FM-to-MC transition. Limitations associated with data acquisition via the drop shape apparatus, namely the difference between polymer concentration at the chloroform-water and core-corona interfaces, and the model's reliance on long chain statistics prevents a full quantitative comparison from being provided within the scope of this manuscript. We intend to continue to develop models in collaboration with the Dr. Shull to further understand and enhance FM-to-MC delivery systems in future publications.

5. Please provide size distribution histogram analysis of spherical micelles observed in cryogenic TEM micrographs to corroborate results from dynamic light scattering (DLS) experiments.

Additional cryoTEM micrographs were acquired of MCs in the supernatants of scaffolds irradiated for 6 and 24 hours. For each scaffold formulation at each time point, three separate micrographs

were captured at an intermediate magnification (4,000 x nominal magnification). A total of 500 individual MCs were manually sized via ImageJ to assess micellar size distributions for each formulation at both the 6 and 24 hour timepoints. It should be noted that due to the limited contrast and, in some instances, tight packing of individual PEG-*bl*-PPS MCs, automated counting was not applicable. The generated size distribution histograms have been overlaid with a representative micrograph in Figure 4b.

6. Figure 2a. Authors should reintroduce all building blocks that are used in this schematic. Please be more specific when mentioning “oxidation”. How is oxidation triggered in this case (photo-oxidation *via* ethyl eosin upon white light irradiation)?

We thank the reviewer for pointing out this lack of detail. The figure caption has been updated with the following text:

Fig. 3b: “Graphical depiction of an FM-scaffold crosslinking with 8-arm PEG-thiol and subsequent oxidation-triggered induction of the cylinder-to-sphere transition for release of micelles (MCs). FMs with PPS cores (blue) and PEG outer coronas (green) are shown as networks that can be crosslinked into stable porous scaffolds through modular incorporation of thiol (red) reactive BCP end-functionalized with VS (black) moieties. Oxidation *via* photo-oxidation (*in vitro*) or through physiologic levels of ROS (*in vivo*) induced cylinder-to-sphere (FM-to-MC) transitions for the release of monodisperse micelles.”

7. Page 5, lines 1-3: “Scaffolds exhibited frequency dependence in both their storage and loss moduli at higher frequencies, and the inverse linear dependence of complex viscosity with regard to frequency was indicative of a solid-to-liquid transition (Supplementary Fig. 3c,d).” Please double-check Supplementary Fig. 3c,d. This seems to be mislabeled and should rather read Supplementary Fig. 2c,d.

Thank you for catching this error. The manuscript text has been updated accordingly.

“Scaffolds exhibited frequency dependence in both their storage and loss moduli at higher frequencies, and the inverse linear dependence of complex viscosity with regard to frequency was indicative of a solid-to-liquid transition (**Supplementary Fig. 3c,d**)”

8. Authors indicate on page 5 that they used 0.75% ethyl eosin by mass. Please provide rationale for this number. Has the amount of ethyl eosin that is incorporated into FM been optimized?
A similar point was raised by Reviewer 1 in Comment 4. Please consult the above response for more details. In brief, we have previously optimized and published the use of ethyl eosin payloads for photo-oxidation of PEG-*bl*-PPS nanocarriers (Vasdekis & Scott *et al*, ACS Nano 2012). Building off this previous work, we explored ethyl eosin concentrations ranging from 0.25% - 0.75% ethyl eosin by mass (Supplementary Fig. 3e,f). After determining that there was not a significant difference in the encapsulation efficiency at these three concentrations, we utilized 0.75% ethyl eosin as it resulted in the largest encapsulated payload.
9. Page 5, lines 14-16: “CryoTEM and dynamic light scattering (DLS) were conducted on the supernatant surrounding the irradiated scaffolds, revealing monodisperse populations of spherical

micelles despite varying percentages of VS-BCP (Fig. 3a,b).” Please provide size distribution histogram analysis of cryoTEM micrographs to corroborate DLS results.

CryoTEM micrographs have been used to generate size distributions to corroborate DLS analysis. Generated histograms have been overlaid with a representative image in Figure 4b. As noted in our response to Question 7 posed by Reviewer 1, discrepancies between the DLS analysis and cryoTEM analysis can be explained by the lack of contrast exhibited by the PEG corona. Further characterization of FMs was provided by SAXS analysis presented in Figure 1c.

10. What temperature was used for *in vitro* photoinduced oxidation of FM scaffolds? How many nanoparticles (MCs) were released for various FM scaffolds over time?

Photodegradation studies conducted *in vitro* were completed at room temperature. The objective was to achieve more rapid FM-to-MC transitions for better characterization and analysis of oxidation-dependent MC release. Text was added to the results and discussion sections to further clarify this.

Reviewer 2 poses a great question concerning the quantification of micellar release. Initially, we planned to assess this very characteristic as well. But due to the size and composition of the MCs, we cannot accurately quantify their concentration within the supernatant. Instruments, such as Malvern’s NanoSight, offer the ability to estimate nanoparticle concentration. But the lower detection limit for soft, polymeric materials is between 30 – 40 nm. As such, the concentration of PEG-*bl*-PPS MCs cannot accurately be quantified. We believe the increase in absorbance within the supernatant following scaffold irradiation (depicted in Supplementary Figure 4c) successfully conveys that scaffold oxidation results in scaffold mass loss that coincides with an increase in PEG-*bl*-PPS within the supernatant. The combination of absorbance measurements and cryoTEM suggest that the absorbance measurements can be used as a relative assessment of changes in nanoparticle concentration.

11. Page 7, lines 20-22: “CryoTEM confirmed that incorporation of the DyLight-conjugated BCPs into the FMs did not alter FM morphology (Supplementary Fig. 5).” Supplementary Fig. 5 does not provide cryoTEM micrographs.

Thank you for catching this error. The text has been corrected.

12. Please provide images for intravital fluorescence imaging results shown in Figure 4b. These images can be put into the Supplementary Information. Please explain in detail how images were processed to obtain these results.

All intravital fluorescence images corresponding to revised Figure 5b have been placed in Supplementary Figure 6. Those corresponding to revised Figure 6 have been placed in Supplementary Figure 7. The follow explanation has been added to the Methods section to provide detail on how images were analyzed:

Page 26: “To process images, all timepoints corresponding to a single treatment group were simultaneously loaded into Living Image software. Visualization of DyLight signal was scaled per treatment rather than individual mouse. The minimum threshold value for signal visualization was

increased until signal depicted on the feet and tails of all mice in the analysis was removed. Circular ROIs were applied for each mouse in the treatment group and adjusted to an area that encompassed all visible signal. Size adjusted ROIs were generated across timepoints for individual mice allowing for equivalent ROIs to be applied across all timepoints within the study. Total radiant efficiency was measured and recorded. The average background signal, recorded in an untreated A/J mouse, was used to calculate the total background radiant efficiency for each ROI. The total radiant efficiency associated with only the presence of DyLight-755 was calculated by subtracting the background radiant efficiency from the total radiant efficiency as measured in Living Image software.”

13. Figure 4. Please include a schematic that illustrates the experimental design for FM scaffold injection condition and control condition, *i.e.*, DiI group. Is not clear why CD45+ cells should be double positive (DiI+ and DyLight633+) when only free solubilized DiI in PBS was injected. Where does DyLight633 signal come from, if only DiI was injected? This should be clarified.

While we greatly appreciate the reviewer’s comment, we feel that Fig. 5d and 5e already show that we are comparing an *in situ* formed scaffold group with a free DiI injection group for this experiment. Reviewer 2 makes an excellent point in that our current gating strategy could lead to some confusion amongst readers due to cell autofluorescence. We have therefore regated the data to account for cell autofluorescence. Figure 5d and Figure 5e in the revised manuscript and Supplementary Figure 12 depict the changes in gating strategy. In Figure 5d, the percentage of double positive and DyLight-633+ cells in the representative DiI control have been reduced from 0.04% and 1.14% to 0.01% and 0.04%, respectively. As a consequence of this change in gating strategy, the percentage of double positive cells in the scaffold treated example decreased from 3.48% to 1.59%, but the adjusted r^2 value from Pearson’s correlation coefficient derived from the linear fit of the events within the double positive quadrant of the scaffold treated example increased from 0.9355 to 0.9773. Similar changes manifest themselves in Figure 5e in comparison to what was previously Figure 4e and said changes led to an increase in statistical significance between the control and scaffold groups for both CD45+ and F4/80+ cells.

14. Page 13, lines 3-6: “Uptake within the inguinal lymph nodes and liver was not statistically significant from background (Supplementary Fig. 6b). Comparison of H&E and Masson’s Trichrome stained tissue sections indicate only a mild increase in collagen deposition and macrophage infiltration for the mice receiving FM-scaffolds (Fig. 5d-i, Supplementary Fig.7).” Please double-check Supplementary Figures. The labeling does not match. For example, Supplementary Fig. 6b is an histology image of uncrosslinked FM.

The revised manuscript has been updated accordingly.

“Uptake within the inguinal lymph nodes and liver was not statistically significant from background (Supplementary Fig. 9). Comparison of H&E and Masson’s Trichrome stained tissue sections indicate only a mild increase in collagen deposition and macrophage infiltration for the mice receiving FM-scaffolds (Fig. 6d-i, Supplementary Fig.10).”

15. Supplementary Figure 6. Control group is missing that did not receive any injection.

We believe that the Free DyLight control is an effective comparison and makes an uninjected control redundant and an unnecessary use of animals. Our interest is in determining the extent of fibrous capsule formation in the FM-scaffold treated mice, and the bolus free DyLight injection, which can effectively be drained from the subcutaneous space via the lymphatics, does not induce this physiologic response and is thus a sufficient control. Therefore, euthanization of untreated control mice in this instance could be considered surplus to what is required.

16. Page 11, line 21-24 and page 13, line 1: “Specifically, MHCII- dendritic cells and macrophages exhibited a discernible increase in MC fluorescence in comparison to free FM and DyLight controls. A statistically significant increase in MC fluorescence was also observed within MHCII+ dendritic cells when comparing mice receiving *in situ* formed FM-scaffolds in comparison to the free DyLight control.” Please discuss in more detail the underlying mechanisms for increased MC fluorescence signal in MHCII- dendritic cells and macrophages.

Reviewer 2 makes an excellent suggestion in expounding upon why an increased uptake was observed in MHCII- dendritic cells and macrophages. Dendritic cells (DCs), macrophages, and B cells, compose a classification of immune cells known as professional antigen presenting cells (APCs). APCs, which are highly phagocytic and central to the mononuclear phagocyte system, internalize foreign material, process antigen, and present antigen to T cells for activation. This ability to activate T cells coupled with their potency for cytokine release make them central figures in dictating the body’s immune response (Scott *et al*, Annu Rev Biomed Eng 2017). Of the professional APCs, DCs and macrophages exhibit the greatest phagocytic potential and as such, are expected to show increased fluorescence if in fact micellar structures are present. When looking at DCs, MHCII expression is often used to assess DC maturation and it has been shown that mature DCs exhibit reduced uptake. Therefore, the statistically significant increase in nanoparticle MFI within DCs, MHCII- DCs, and macrophages is logical given the phagocytic capacity of these cells. Text was added to the discussion section to highlight the relevance of these cells for nanocarrier uptake.

17. For *in vitro* MC release studies, authors used photoinduced oxidation based on white light illumination of ethyl eosin that was incorporated into FM scaffolds. In *in vivo* studies, authors did not incorporate ethyl eosin into the FM scaffolds and there was also no external trigger for MC release, such as white light illumination in the case of *in vitro* studies. What is the mechanism for MC release from FM scaffolds *in vivo*? How can the MC *in vivo* release be controlled? Do kinetics and efficiencies of MC *in vivo* release get altered when healthy and diseased mice are compared, *i.e.*, does the presence of a disease change the physiological oxidation of FM scaffolds to release MCs? If yes, how does this change cellular interaction of MCs *in vivo*?

These are great points posed by Reviewer 2 and indicate the broad interest and wide range of applications and future studies that could result from this manuscript. We feel that almost all of these questions are better addressed in separate focused studies that we and others can perform following publication of the current manuscript.

The *in vivo* application of FM-scaffolds relies on the presence of physiologic concentrations of reactive oxygen species (ROS) to trigger the FM-to-MC transition and subsequent micellar release. A variety of biologically relevant ROS exist, including H₂O₂, superoxide anion, and hydroxyl radicals, and their presence is the result of both endogenous sources, such as through mitochondrial

oxidative phosphorylation, and exogenous sources (Ray *et al*, Cellular signaling 2012). Under homeostasis, ROS plays a role as a signaling molecule (Ray *et al*, Cellular signaling 2012), but it also plays a central role in the phagocytic response of innate immune cells via oxidative burst (West *et al*, Nature 2011). While difficult to quantify, biologically relevant concentrations of H₂O₂ have been estimated to range from 50 – 100 μM (de Gracia Lux *et al*, JACS 2012) and our work suggests that physiologic concentrations of ROS, even under homeostasis, coupled with the oxidative burst provided by innate immune cells responding to the presence of the FM-scaffold are sufficient to induce the FM-to-MC transition *in vivo*. We have edited the text to clarify these points.

The following text was added to the results and discussion sections for clarity:

Page 11: “While difficult to quantify, biologically relevant concentrations of reactive oxygen species (ROS) have been estimated to range from 50 – 100 μM⁶², and we have previously demonstrated that H₂O₂ at as low as 5 μM can induce changes in PEG-bl-PPS nanocarrier morphology⁶. We therefore hypothesized that continuous exposure of FM-scaffolds to physiologic levels of oxidation could be sufficient to induce sustained FM-to-MC transitions *in vivo* following subcutaneous injection in mice.”

Page 15: “Following subcutaneous injection and *in situ* gelation in mice, our results suggest, although indirectly, that physiologic concentrations of ROS under homeostatic conditions are sufficient to induce the FM-to-MC transition *in vivo* for sustained release of nanocarriers.”

Application of the FM-scaffold specifically to disease states associated with elevated concentrations of ROS, such as cancer, or inflammation-related pathologies, like peripheral arterial disease, would most likely alter the oxidation kinetics of the propylene sulfide and therefore the release rate of MCs. While kinetics would be altered, individual interactions between MCs and phagocytic cells would not be expected to significantly differ.

Reviewer 2 asks a great question concerning how one could use an external trigger to control scaffold degradation. We are currently working on PEG-bl-PPS derivatives that would permit the use of either near infrared fluorescence or ultrasound to control singlet oxygen generation to induce the FM-to-MC transition *in vivo*. We envision the synthesis and optimization of these on-demand *in vivo* platforms to be more suitable as future separate standalone manuscripts.

18. The manuscript is missing an application/proof-of-concept which strongly limits its significance and broad interest. Authors show that MCs can be released *in vivo* from FM scaffolds and that these MCs can interact with immune cells. However, it is not clear why this would be a beneficial approach. Authors need to demonstrate in a proof-of-concept study that their approach is useful in modulating biological processes/physiological conditions.

We agree with the reviewer and are planning several different studies that employ this FM-to-MC sustained delivery system. But the focus of this manuscript is on characterizing and demonstrating a novel form of controlled release. Including a proof-of-concept study that highlights the platform’s ability to modulate a biologic problem would, in our opinion, make the manuscript too convoluted and unfocused, shifting the focus away from our goal of demonstrating that the FM-to-MC transition can be used for *in vivo* delivery. We also disagree with the assessment that the short-term (one week) delivery of a model hydrophobic molecule (DiI) and long-term (one month) release from the depot site are not of significance to a broad scientific community. This manuscript provides a novel alternative to current hydrogel-nanoparticle composite delivery systems, where

the hydrogel network plays a role in modulating nanoparticle release but plays no direct role in delivery of the active. We employ an extensive analysis of the FM-to MC transition using diverse electron microscopy techniques, SAXS and thermodynamic models. Furthermore, we believe that the sustained *in vivo* release of intact micellar delivery vehicles enhances the appeal of this work. Nanoparticles have proven to be effective delivery agents to professional APCs. Given the broad range of biologic applications in which the sustained modulation of professional APCs is of interest (cancer immunotherapy, subunit vaccine development, modulation of cardiovascular disease, diabetes, and various autoimmune diseases), the work pertaining to this manuscript should appeal across the biology community (Scott *et al*, Annu Rev Biomed Eng 2017). In addition to the broad appeal across the field of biology, the chemical, physical, and theoretical tools utilized to acquire data for this manuscript should appeal to members within the fields of chemistry, material science, polymer physics, and nanotechnology.

19. In summary, the manuscript does not meet the level of *Nature Communications* in terms of significance, broad interest, and scholarly presentation. Publication of the manuscript in *Nature Communications* is not recommended. The manuscript is not likely to be one of the five most significant papers in the discipline this year. The strategy described in this manuscript is not sufficiently promising to encourage resubmission to this journal.

In addition to our response to Question 18 above, we hope that with our recent edits, inclusion of a thermodynamic model, and additional electron microscopy, our manuscript now meets the reviewer's standards for significance, broad interest and scholarly presentation.

Referee 3:

1. Supplementary Figure 4 is incorrectly called out in the text as Suppl Fig. 5.

Thank you for catching this error. We have updated the manuscript so that it correctly lists this figure within the manuscript.

2. The free dye control case reported in Supplementary Fig. 5 is confusing: If there is not Alexa633-labeled block copolymer present in the free dye case, how can there be a double+ population of cells? If the gating were rigorously excluding autofluorescence, this population should be by definition zero. This analysis should be revisited/clarified.

Reviewer 3 correctly pointed out that our previous gating strategy could cause confusion among our audience. We have rigorously gated this data to exclude the DyLight-633 autofluorescence, which should be absent in the DiI receiving control mice. A detailed explanation of the changes that manifested following this adjustment can be found under the response to Reviewer 2 Question 13.

3. It would be useful for the authors to report on the % of nanoparticle+ cells to accompany Fig. 5c, or show raw histograms of micelle signal in the different cell populations in supplemental, to give the reader a better sense of how much material is still present at the late time point shown.

As requested, raw histograms of the micelle signal within different immune cell populations have been generated and included in Supplementary Figure 8.

4. Why does dye release plateau for both the uncrosslinked and crosslinked scaffold groups at ~75% released, rather than steadily continuing on toward 100% release at late times? Could this reflect that ~25% is being phagocytosed by sessile macrophages at the injection site? Or a portion walled off by the host response and unable to exit the tissue? This is an important point that should be clarified.

The points posed by Reviewer 3 are valid concerns and require additional discussion within the text of the manuscript. Given the results observed in the histological analysis provided in Figure 6d-i and Supplementary Figure 10, it is unlikely that ~20% of fluorescent signal that remains is due to fibrous capsule formation given the minimal increase in collagen deposition at the injection site. While the release rate has significantly decreased in comparison to what was observed at the early stages of the experiment, an ~5% decrease in fluorescence signal was still observed for both groups over the last week of the study (0.71% released/day). It is possible that this change in the release rate is simply due to the reduction in total material at the injection site.

For clarity, the following text was added to the results section:

Page 15: “This observed lack of an inflammatory response suggests that the gradual decrease in the release rate observed in Figure 6b is not due to walling-off of the scaffold by fibrous capsule formation and may instead simply reflect the reduction in total material at the injection site over time.”

REVIEWERS' COMMENTS:

Reviewer #1 (Remarks to the Author):

This Reviewer is very pleased with the answers provided by the author to all the concerns. Each point was addressed with extensive explanation, modification of the manuscript, and even additional experiments.

This reviewer considers the paper completely appropriate for publication in Nature Communications in the present form.

Reviewer #2 (Remarks to the Author):

Authors have significantly improved the quality of their work. Reviewer comments have been sufficiently addressed. Publication of this manuscript is recommended.

Reviewer #3 (Remarks to the Author):

The authors have made numerous improvements to the manuscript and a commendable effort to respond to the referee comments. The new analysis/modeling added from new co-author Shull is a very useful addition to the manuscript.